# 3DIS: Depth-Driven Decoupled Image Synthesis for Universal Multi-Instance Generation

**Dewei Zhou**[†1], **Ji Xie**[†1], **Zongxin Yang**[†2], **Yi Yang** [*1]

[1]RELER, CCAI, Zhejiang University [2]DBMI, HMS, Harvard University
{zdw1999,sanaka87,yangyics}@zju.edu.cn
{Zongxin_Yang}@hms.harvard.edu
[†] Equal contribution [*] Corresponding author

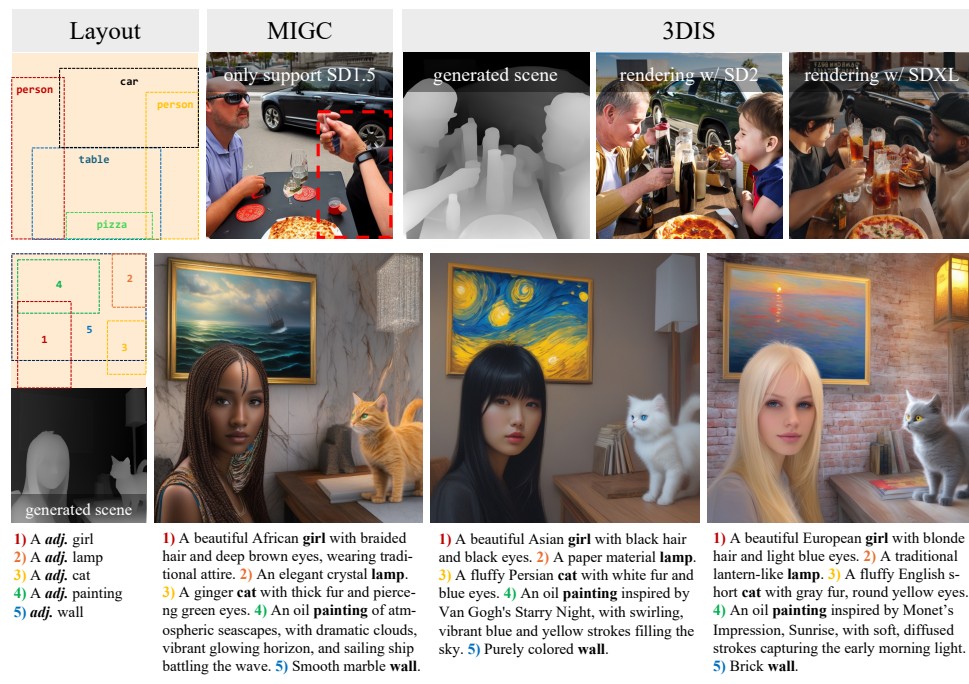

Figure 1: **Images generated using our 3DIS.** Based on the user-provided layout, 3DIS generates a scene depth map that precisely positions each instance and renders their fine-grained attributes without the need for additional training, using a variety of foundational models.

## ABSTRACT

The increasing demand for controllable outputs in text-to-image generation has spurred advancements in multi-instance generation (MIG), allowing users to define both instance layouts and attributes. However, unlike image-conditional generation methods such as ControlNet, MIG techniques have not been widely adopted in state-of-the-art models like SD2 and SDXL, primarily due to the challenge of building robust renderers that simultaneously handle instance positioning and attribute rendering. In this paper, we introduce **D**epth-**D**riven **D**ecoupled **I**mage **S**ynthesis (3DIS), a novel framework that decouples the MIG process into two stages: (i) generating a coarse scene depth map for accurate instance positioning and scene composition, and (ii) rendering fine-grained attributes using pre-trained ControlNet on any foundational model, without additional training. Our 3DIS framework integrates a custom adapter into LDM3D for precise depth-based layouts and employs a finetuning-free method for enhanced instance-level attribute rendering. Extensive experiments on COCO-Position and COCO-MIG benchmarks demonstrate that 3DIS significantly outperforms existing methods in both layout precision and attribute rendering. Notably, 3DIS offers seamless compatibility with diverse foundational models, providing a robust, adaptable solution for advanced multi-instance generation. The code is available at: https://github.com/limuloo/3DIS.

# 1    INTRODUCTION

With the rapid advancement of text-to-image generation technologies, there is a growing interest in achieving more controllable outputs, which are now widely utilized in artistic creation (Zhou et al., 2024c; Zhuo et al., 2024): *(i) Image-conditional generation techniques*, *e.g.*, ControlNet (Zhang et al., 2023), allow users to generate images based on inputs like depth maps or sketches. *(ii) Multi-instance generation (MIG) methods*, *e.g.*, GLIGEN (Li et al., 2023c) and MIGC (Zhou et al., 2024a), enable users to define layouts and detailed attributes for each instance within the generated images.

However, despite the importance of MIG in controllable generation, these methods have not been widely adopted across popular foundational models like SD2 (Rombach et al., 2023) and SDXL (Podell et al., 2023), unlike the more widely integrated ControlNet. Current state-of-the-art MIG methods mainly rely on the less capable SD1.5 (Rombach et al., 2022) model.

We argue that the limited adoption of MIG methods is not merely due to **resource constraints** but also stems from a more fundamental challenge, *i.e.*, **unified adapter challenge**. Current MIG approaches train a single adapter to handle both instance positioning and attribute rendering. This unified structure complicates the development of robust renderers for fine-grained attribute details, as it requires large amounts of high-quality instance-level annotations. These detailed annotations are more challenging to collect compared to the types of controls used in image-conditional generation, such as depth maps or sketches.

To address the unified adapter challenge and enable the use of a broader range of foundational models for MIG, we propose a novel framework called **D**epth-**D**riven **D**ecoupled **I**mage **S**ynthesis (3DIS). 3DIS tackles this challenge by decoupling the image generation process into two distinct stages, as shown in Fig. 2. *(i) Generating a coarse scene depth map*: During this stage, the MIG adapter ensures accurate instance positioning, coarse attribute alignment, and overall scene harmony without the complexity of fine attribute rendering. *(ii) Rendering a fine-grained RGB image*: Based on the generated scene depth map, we design a finetuning-free method that leverages any popular foundational model with pretrained ControlNet to guide the overall image generation, focusing on detailed instance rendering. This approach *requires only **a single training process*** for the adapter at stage *(i)*, enabling **seamless integration with different foundational models** without needing retraining for each new model.

The 3DIS architecture comprises three key components: *(i) Scene Depth Map Generation*: We developed the first layout-controllable text-to-depth generation model by integrating a well-designed adapter into LDM3D (Stan et al., 2023). This integration facilitates the generation of precise, depth-informed layouts based on instance conditions. *(ii) Layout Control*: We introduce a method to leverage pretrained ControlNet for seamless integration of the generated scene depth map into the generative process. By filtering out high-frequency information from ControlNet's feature maps, we enhance the integration of low-frequency global scene semantics, thereby improving the coherence and visual appeal of the generated images. *(iii) Detail Rendering*: Our method performs Cross-Attention operations separately for each instance to achieve precise rendering of specific attributes (*e.g.*, category, color, texture) while avoiding attribute leakage. Additionally, we use SAM for semantic segmentation on the scene depth map, optimizing instance localization and resolving conflicts from overlapping bounding boxes. This advanced approach significantly improves the rendering of detailed and accurate multi-instance images.

We conducted extensive experiments on two benchmarks to evaluate the performance of 3DIS: *(i) COCO-Position* (Lin et al., 2015; Zhou et al., 2024a): Evaluated the layout accuracy and coarse-grained category attributes of the scene depth maps. *(ii) COCO-MIG* (Zhou et al., 2024a): Assessed the fine-grained rendering capabilities. The results indicate that 3DIS excels in creating superior scenes while preserving the accuracy of fine-grained attributes during detailed rendering. On the COCO-Position benchmark, 3DIS achieved a **16.3**% improvement in $AP_{75}$ compared to the previous state-of-the-art method, MIGC. On the COCO-MIG benchmark, our training-free detail rendering approach improved the Instance Attribute Success Ratio by **35**% over the training-free method Multi-Diffusion (Bar-Tal et al., 2023) and by **5.5**% over the adapter-based method InstanceDiffusion (Wang et al., 2024). Furthermore, the 3DIS framework can be seamlessly integrated with off-the-shelf adapters like GLIGEN and MIGC, thereby enhancing their rendering capabilities.

In summary, the key contributions of this paper are as follows:

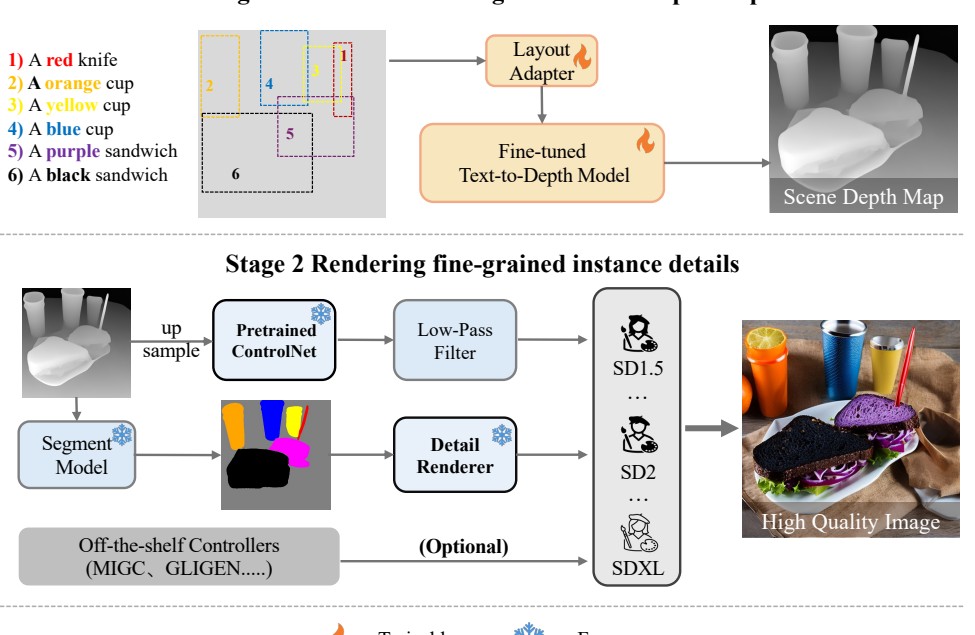

Figure 2: **The overview of 3DIS.** 3DIS decouples image generation into two stages: creating a scene depth map and rendering high-quality RGB images with various generative models. It first trains a Layout-to-Depth model to generate a scene depth map. Then, it uses a pre-trained ControlNet to inject depth information into various generative models, controlling scene representation. Finally, a training-free detail renderer renders the fine-grained attributes of each instance.

- We propose a novel **3DIS framework** that decouples multi-instance generation into two stages: adapter-controlled scene depth map generation and training-free fine-grained attribute rendering, enabling integration with various foundational models.
- We introduce the first **layout-to-depth model** for multi-instance generation, which improves scene composition and instance positioning compared to traditional layout-to-RGB methods.
- Our **training-free detail renderer** enhances fine-grained instance rendering without additional training, significantly outperforming state-of-the-art methods while maintaining compatibility with pretrained models and adapters.

## 2 RELATED WORK

**Controllable Text-to-Image Generation.** With the rapid advancements in diffusion models (Zhou et al., 2023; 2025; Lu et al., 2023; 2024a;b; Zhao et al., 2024a;b; Gao et al., 2024; Xie et al., 2024) and multimodal learning (Wang et al., 2025; Yang et al., 2021; Li et al., 2023b; Yang et al., 2024b), text-to-image technologies (Rombach et al., 2022; Podell et al., 2023) have reached a level where they can produce high-quality images. Researchers are now increasingly focused on enhancing their control over the generated content. Numerous approaches have been developed to improve this control. ControlNet (Zhang et al., 2023) incorporates user inputs such as depth maps and edge maps by training an additional side network, allowing for precise layout control in image generation. Methods like IPAdapter (Ye et al., 2023) and PhotoMaker (Li et al., 2024) generate corresponding images based on user-provided portraits. Techniques such as ELITE (Wei et al., 2023) and SSR-Encoder (Zhang et al., 2024) enable networks to accept specific conceptual image inputs for better customization. Additionally, MIGC (Zhou et al., 2024b) and InstanceDiffusion (Wang et al., 2024) allow networks to generate images based on user-specified layouts and instance attribute descriptions, defining this task as Multi-Instance Generation (MIG), which is the focal point of this paper.

**Multi-Instance Generation (MIG).** MIG involves generating each instance based on a given layout and detailed attribute descriptions, while maintaining overall image harmony. Current MIG methods

primarily use Stable Diffusion (SD) architectures, classified into three categories: 1) Training-free methods: Techniques like BoxDiffusion (Xie et al., 2023) and RB (Xiao et al., 2023) apply energy functions to attention maps, enabling zero-shot layout control by converting spatial guidance into gradient inputs. Similarly, Multi-Diffusion (Bar-Tal et al., 2023) generates instances separately and then combines them according to user-defined spatial cues, enhancing control over orientation and arrangement. 2) Adapter methods: Approaches like GLIGEN (Li et al., 2023c) and InstanceD-iffusion (Wang et al., 2024) integrate trainable gated self-attention layers (Vaswani et al., 2017; Yang et al., 2024c; 2025) into the U-Net (Ronneberger et al., 2015), improving layout assimilation and instance fidelity. MIGC (Zhou et al., 2024a;b) further divides the task, using an enhanced attention mechanism to generate each instance precisely before integration. 3) SD-tuning methods: Reco (Yang et al., 2023) and Ranni (Feng et al., 2024) add instance position data to text inputs and fine-tune both CLIP and U-Net, allowing the network to utilize positional cues for more precise image synthesis. Previous methods entangled instance positioning with attribute rendering, complicating the training of a robust instance renderer. Our approach decouples this process into adapter-controlled scene depth map generation and training-free detail rendering. This separation allows the adapter to only handle instance positioning and coarse attributes, while leveraging the generative priors of pre-trained models, enhancing both flexibility and performance.

## 3 METHOD

### 3.1 PRELIMINARIES

Latent Diffusion Models (LDMs) are among the most widely used text-to-image models today. They significantly enhance generation speed by placing the diffusion process for image synthesis within a compressed variational autoencoder (VAE) latent space. To ensure that the generated images align with user-provided text descriptions, LDMs typically employ a Cross Attention mechanism, which integrates textual information into the image features of the network. In mathematical terms, the Cross Attention operation can be expressed as follows:

$$\text{Attention}(\mathbf{Q}, \mathbf{K}, \mathbf{V}) = \text{Softmax}\left(\frac{\mathbf{Q}\mathbf{K}^\top}{\sqrt{d_k}}\right)\mathbf{V}, \tag{1}$$

where $\mathbf{Q}$, $\mathbf{K}$, and $\mathbf{V}$ represent the query, key, and value matrices derived from the image and text features, respectively, while $d_k$ denotes the dimension of the key vectors.

### 3.2 OVERVIEW

Fig. 2 illustrates the overview framework of the proposed 3DIS, which decouples image generation into coarse-grained scene construction and fine-grained detail rendering. The specific implementation of 3DIS consists of three steps: **1) Scene Depth Map Generation (§ 3.3)**, which produces a corresponding scene depth map based on the user-provided layout; **2) Global Scene Control (§ 3.4)**, which ensures that the generated images align with the scene maps, guaranteeing that each instance is represented; **3) Detail Rendering (§ 3.5)**, which ensures that each generated instance adheres to the fine-grained attributes described by the user.

### 3.3 SCENE DEPTH MAP GENERATION

In this section, we discuss how to generate a corresponding depth map based on the user-provided layout, creating a coherent and well-structured scene while accurately placing each instance.

**Choosing the text-to-depth model.** Upon investigation, we identified RichDreamer (Qiu et al., 2024) and LDM3D (Stan et al., 2023) as the primary models for text-to-depth generation. Rich-Dreamer fine-tunes the pretrained RGB Stable Diffusion (SD) model to generate 3D information, specifically depth and normal maps, while LDM3D enables SD to produce both RGB images and depth maps simultaneously. Experimental comparisons show LDM3D outperforms RichDreamer in complex scenes, likely due to its concurrent RGB and depth map generation. This dual capability preserves RGB image quality while enhancing depth map generation, making LDM3D our preferred model for text-to-depth generation.

**Fine-tuning the text-to-depth model.** In contrast to RGB images, depth maps typically prioritize the restoration of low-frequency components over high-frequency details. For instance, while

a texture-rich skirt requires intricate details for RGB image generation, its corresponding depth map remains relatively smooth. Therefore, we aim to enhance the model's ability to recover low-frequency content. Low-frequency components often indicate significant redundancy among adjacent pixels. To simulate this characteristic, we implemented an augmented pyramid noise strategy (Kasiopy, 2023), which involves downsampling and then upsampling randomly sampled noise $\epsilon$ to create patterns with high redundancy between adjacent pixels. We used the original SD training loss (Rombach et al., 2022) to fine-tune our text-to-depth model $\theta$, but adjusted the model to predict this patterned noise $\epsilon_{\text{pyramid}}$ with the text prompt $c$:

$$\min_{\theta} \mathcal{L}_{\text{text}} = \mathbb{E}_{z,\epsilon \sim \mathcal{N}(0,I),t} \left[ \left\| \epsilon_{\text{pyramid}} - f_{\theta}(z_t, t, c) \right\|_2^2 \right]. \tag{2}$$

**Training the Layout-to-depth adapter.** Similar to previous methodologies (Zhou et al., 2024a; Li et al., 2023c; Wang et al., 2024), we incorporated an adapter into our fine-tuned text-to-depth model, enabling layout-to-depth generation, specifically leveraging the state-of-the-art MIGC (Zhou et al., 2024a) model. Unlike earlier approaches, our method for generating depth maps does not rely on detailed descriptions of specific instance attributes, such as material or color. Consequently, we have augmented the dataset used for MIGC by eliminating fine-grained attribute descriptions from the instance data, thus focusing more on the structural properties of individual instances and the overall scene composition. The training process for the adapter $\theta'$ can be expressed as:

$$\min_{\theta'} \mathcal{L}_{\text{layout}} = \mathbb{E}_{z,\epsilon \sim \mathcal{N}(0,I),t} \left[ \left\| \epsilon_{\text{pyramid}} - f_{\theta,\theta'}(z_t, t, c, l) \right\|_2^2 \right], \tag{3}$$

where the base text-to-depth model $\theta$ is frozen, and the $l$ is the input layout.

## 3.4 GLOBAL SCENE CONTROL

In this section, we will describe how to control the generated images to align with the layout of the generated scene depth map, ensuring that each instance appears in its designated position.

**Injecting depth maps with ControlNet.** After generating scene depth maps with our layout-to-depth models, we employed the widely adopted ControlNet (Zhang et al., 2023) model to incorporate global scene information. Scene depth maps focus on overall scene structure, without requiring fine-grained detail. Thus, although the base model produces 512x512 resolution maps, they can be upsampled to 768x768, 1024x1024, or higher (see Fig. 3 and Fig. 4, e.g., SD2 and SDXL). Since most generative models have depth ControlNet versions, these maps can be applied across various models, ensuring accurate instance placement and mitigating omission issues.

**Removing high-frequency noise in depth maps.** In our framework, the injected depth maps are designed to manage the low-frequency components of the constructed scene, while the generation of high-frequency details is handled by advanced grounded text-to-image models. To enhance the integration of these components, we implement a filtering process to remove high-frequency noise from the feature maps generated by ControlNet before injecting them into the image generation network. Specifically, the scene condition feature output from ControlNet, denoted as $F$, is added to the generation network. Prior to this addition, we transform $F$ into the frequency domain via the Fast Fourier Transform (FFT) and apply a filter to attenuate the high-frequency components:

$$F_{\text{filtered}} = \mathcal{F}^{-1}\left(H_{\text{low}} \cdot \mathcal{F}(F)\right), \tag{4}$$

where $\mathcal{F}$ and $\mathcal{F}^{-1}$ denote the FFT and inverse FFT, respectively, and $H_{\text{low}}$ represents a low-pass filter applied in the frequency domain. This approach has been shown to reduce the occurrence of artifacts and improve the overall quality of the generated images without reducing performance.

## 3.5 DETAILS RENDERING

Through the control provided by ControlNet, we can ensure that the output images align with our generated scene depth maps, thus guaranteeing that each instance appears at its designated location. However, we still lack assurance regarding the accuracy of attributes such as category, color, and material for each instance. To render each instance with correct attributes, we propose a training-free **detail renderer** to replace the original Cross-Attention Layers for this purpose. The process of rendering an entire scene using a detail renderer can be broken down into the following three steps.

Table 1: **Quantitative results on COCO-Position (§4.3).** We only utilize complex layouts that contain at least five instances, resulting in significant overlap.

| Method | Layout Accuracy | | | Instance Accuracy | | | Image Quality | |
|---|---|---|---|---|---|---|---|---|
| | $AP\uparrow$ | $AP_{75}\uparrow$ | $AP_{50}\uparrow$ | $SR_{inst}\uparrow$ | MIoU | $CLIP\uparrow$ | $SR_{img}\uparrow$ | $FID\downarrow$ |
| BoxDiff [ICCV23] | 3.15 | 2.12 | 10.92 | 22.74 | 27.28 | 18.82 | 0.53 | 25.15 |
| MultiDiff [ICML23] | 6.37 | 4.24 | 13.22 | 28.75 | 34.17 | 20.12 | 0.80 | 33.20 |
| GLIGEN [CVPR23] | 38.49 | 40.75 | 63.79 | 83.31 | 70.14 | 19.61 | 40.13 | 26.80 |
| MIGC [CVPR24] | 45.03 | 46.15 | 80.09 | 83.37 | 71.92 | 20.07 | 43.25 | 24.52 |
| 3DIS (SD1.5) | **56.83** | **62.40** | **82.29** | **84.71** | **73.32** | **20.84** | **46.50** | **23.24** |
| vs. prev. SoTA | **+11.8** | **+16.3** | **+2.2** | **+1.3** | **+1.4** | **+0.8** | **+3.3** | **+1.3** |

**Rendering each instance separately.** For an instance $i$, ControlNet ensures that a shape satisfying its descriptive criteria is positioned within the designated bounding box $\boldsymbol{b}_i$. By applying Cross Attention using the text description of the instance $i$, we can ensure that the attention maps generate significant response values within the $\boldsymbol{b}_i$ region, accurately rendering the attributes aligned with the instance's textual description. For each Cross-Attention layer in the foundation models, we independently render each instance $i$ with their text descriptions to obtain the rendered result $\mathbf{r}_i$, while similarly applying the global image description to yield rendering background $\mathbf{r}_c$. Our next step is to merge the obtained feature maps $\{\mathbf{r}_1, \cdots, \mathbf{r}_n, \mathbf{r}_c\}$ into a single feature map, aligning with the forward pass of the original Cross-Attention layers.

**SAM-Enhancing Instance Location.** While mering rendering results, acquiring precise instance locations helps prevent attribute leakage between overlapping bounding boxes and maintains structural consistency with the instances in the scene depth maps. Consequently, we employ the SAM (Kirillov et al., 2023) model to ascertain the exact position of each instance. For an instance $i$, by utilizing our generated scene depth map $\mathbf{m}_{scene}$ alongside its corresponding bounding box $\boldsymbol{b}_i$, we can segment the specific shape mask $\mathbf{m}_i$ of this instance, thereby facilitating subsequent merging:

$$\mathbf{m}_i = \text{SAM}(\mathbf{m}_{scene}, \mathbf{b}_i) \tag{5}$$

**Merging rendering results.** We employ the precise mask $\mathbf{m}_i$ obtained from SAM to constrain the rendering results of instance $i$ to its own region, ensuring no influence on other instances. Specifically, we construct a new mask $\mathbf{m}'_i$ by assigning a value of $\alpha$ to the areas where $\mathbf{m}_i$ equals 1, while setting all other regions to $-\infty$. Simultaneously, we assign a background value of $\beta$ to the global rendering $\mathbf{r}_c$ through a mask $\mathbf{m}'_c$. By applying the softmax function to the set $\{\mathbf{m}'_1, \mathbf{m}'_2, \ldots, \mathbf{m}'_n, \mathbf{m}'_c\}$, we derive the spatial weights $\{\mathbf{m}''_1, \mathbf{m}''_2, \ldots, \mathbf{m}''_n, \mathbf{m}''_c\}$ for each rendering instance. At each Cross Attention layer, the output can be expressed as follows to render the whole scene:

$$\mathbf{r} = \mathbf{m}''_1 \cdot \mathbf{r}_1 + \mathbf{m}''_2 \cdot \mathbf{r}_2 + \ldots + \mathbf{m}''_n \cdot \mathbf{r}_n + \mathbf{m}''_c \cdot \mathbf{r}_c \tag{6}$$

## 4 EXPERIMENT

### 4.1 IMPLEMENT DETAILS

**Tuning of text-to-depth models.** We utilized a training set comprising 5,878 images from the LAION-art dataset (Schuhmann et al., 2021), selecting only those with a resolution exceeding 512x512 pixels and an aesthetic score of $\geq 8.0$. Depth maps for each image were generated using Depth Anything V2 (Yang et al., 2024a). Given the substantial noise present in the text descriptions associated with the images in LAION-art, we chose to produce corresponding image captions using BLIP2 (Li et al., 2023a). We employed pyramid noise (Kasiopy, 2023) to fine-tune the LDM3D model for 2,000 steps, utilizing the AdamW (Kingma & Ba, 2017) optimizer with a constant learning rate of $1e^{-4}$, a weight decay of $1e^{-2}$, and a batch size of 320.

**Training of the layout-to-depth adapter.** We adopted the MIGC (Zhou et al., 2024a) architecture as the adapter for layout control. In alignment with this approach, we utilized the COCO dataset (Lin et al., 2015) for training. We employed Stanza (Qi et al., 2020) to extract each instance description from the corresponding text for every image and used Grounding-DINO (Liu et al., 2023) to obtain the image layout. Furthermore, we augmented each instance's description by incorporating modified versions that omitted adjectives, allowing our layout-to-depth adapter to focus more on global scene

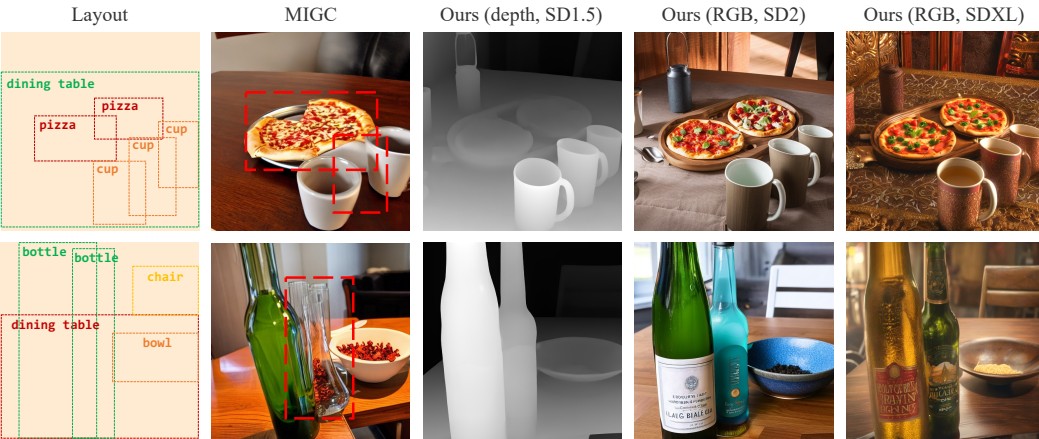

| Layout | MIGC | Ours (depth, SD1.5) | Ours (RGB, SD2) | Ours (RGB, SDXL) |

Figure 3: **Qualitative results on the COCO-Position (§4.3)**.

construction and the coarse-grained categories and structural properties of instances. We maintain the same batch size, learning rate, and other parameters as the previous work.

## 4.2 EXPERIMENT SETUP

**Baselines.** We compared our proposed 3DIS method with state-of-the-art Multi-Instance Generation approaches. The methods involved in the comparison include training-free methods: BoxDiffusion (Xie et al., 2023) and MultiDiffusion (Bar-Tal et al., 2023); and adapter-based methods: GLIGEN (Li et al., 2023c), InstanceDiffusion (Wang et al., 2024), and MIGC (Zhou et al., 2024a).

**Evaluation Benchmarks.** We conducted experiments using two widely adopted benchmarks, COCO-position (Lin et al., 2015) and COCO-MIG (Zhou et al., 2024a), to assess the performance of models in different aspects of instance generation. The COCO-position benchmark emphasizes the evaluation of a model's capacity to control the spatial arrangement of instances, as well as their high-level categorical attributes. In contrast, the COCO-MIG benchmark is designed to test a model's ability to precisely render fine-grained attributes for each generated instance. To rigorously compare the models' performance in handling complex scene layouts, we concentrated our analysis on the COCO-position benchmark, specifically focusing on layouts containing five or more instances. For a comprehensive evaluation, each model generated 750 images across both benchmarks.

**Evaluation Metrics.** We used the following metrics to evaluate the model: *1) Mean Intersection over Union (MIoU)*, measuring the overlap between the generated instance positions and the target positions; *2) Local CLIP score*, assessing the visual consistency of the generated instances with their corresponding textual descriptions; *3) Average Precision (AP)*, evaluating the overlap between the generated image layout and the target layout; *4) Instance Attribute Success Ratio (IASR)*, calculating the proportion of correctly generated instance attributes; *5) Image Success Ratio (ISR)*, measuring the proportion of images in which all instances are correctly generated.

## 4.3 COMPARISON

**Scene Construction.** The results in Tab. 1 demonstrate the superior scene construction capabilities of the proposed 3DIS method compared to previous state-of-the-art approaches. Notably, 3DIS surpasses MIGC with an **11.8**% improvement in AP and a **16.3**% increase in $AP_{75}$, highlighting a closer alignment between the generated layouts and the user input. As shown by the visualizations in Fig. 3, 3DIS achieves marked improvements in scenarios with significant overlap, effectively addressing challenges such as object merging and loss in complex layouts. This results in the generation of a more accurate scene depth map, capturing the global scene structure with greater fidelity.

**Detail Rendering.** The results presented in Tab. 2 demonstrate that the proposed 3DIS method exhibits robust detail-rendering capabilities. Notably, the entire process of rendering instance attributes is **training-free** for 3DIS. Compared to the previous state-of-the-art (SOTA) training-free method, MultiDiffusion, 3DIS achieves a **30**% improvement in the Instance Attribute Success Ratio (IASR). Additionally, when compared with the SOTA adapter-based method, Instance Diffusion,

Table 2: **Quantitative results on proposed COCO-MIG-BOX (§4.3)**. $\mathcal{L}_i$ means that the count of instances needed to generate in the image is **i**.

| | | Instance Attribute Success Ratio↑ | | | | | | Mean Intersection over Union↑ | | | | | |
|---|---|---|---|---|---|---|---|---|---|---|---|---|---|
| Method | | $\mathcal{L}2$ | $\mathcal{L}3$ | $\mathcal{L}4$ | $\mathcal{L}5$ | $\mathcal{L}6$ | $\mathcal{AVG}$ | $\mathcal{L}2$ | $\mathcal{L}3$ | $\mathcal{L}4$ | $\mathcal{L}5$ | $\mathcal{L}6$ | $\mathcal{AVG}$ |
| *Adapter rendering methods* | | | | | | | | | | | | | |
| GLIGEN [CVPR23] | | 41.3 | 33.8 | 31.8 | 27.0 | 29.5 | 31.3 | 33.7 | 27.6 | 25.5 | 21.9 | 23.6 | 25.2 |
| InstanceDiff [CVPR24] | | 61.0 | 52.8 | 52.4 | 45.2 | 48.7 | 50.5 | 53.8 | 45.8 | 44.9 | 37.7 | 40.6 | 43.0 |
| MIGC [CVPR24] | | 74.8 | 66.2 | 67.4 | 65.3 | 66.1 | 67.1 | 63.0 | 54.7 | 55.3 | 52.4 | 53.2 | 54.7 |
| ***training-free** rendering* | | | | | | | | | | | | | |
| TFLCG [WACV24] | | 17.2 | 13.5 | 7.9 | 6.1 | 4.5 | 8.3 | 10.9 | 8.7 | 5.1 | 3.9 | 2.8 | 5.3 |
| BoxDiff [ICCV23] | | 28.4 | 21.4 | 14.0 | 11.9 | 12.8 | 15.7 | 19.1 | 14.6 | 9.4 | 7.9 | 8.5 | 10.6 |
| MultiDiff [ICML23] | | 30.6 | 25.3 | 24.5 | 18.3 | 19.8 | 22.3 | 21.9 | 18.1 | 17.3 | 12.9 | 13.9 | 15.8 |
| 3DIS (SD1.5) | | 65.9 | 56.1 | 55.3 | 45.3 | 47.6 | 53.0 | 56.8 | 48.4 | 49.4 | 40.2 | 41.7 | 44.7 |
| 3DIS (SD2.1) | | 66.1 | 57.5 | 55.1 | 51.7 | 52.9 | 54.7 | 57.1 | 48.6 | 46.8 | 42.9 | 43.4 | 45.7 |
| 3DIS (SDXL) | | 66.1 | 59.3 | 56.2 | 51.7 | 54.1 | 56.0 | 57.0 | 50.0 | 47.8 | 43.1 | 44.6 | 47.0 |
| vs. MultiDiff | | **+35** | **+34** | **+31** | **+33** | **+34** | **+33** | **+35** | **+31** | **+30** | **+30** | **+30** | **+31** |
| *rendering w/ **off-the-shelf** adapters* | | | | | | | | | | | | | |
| 3DIS+GLIGEN | | 49.4 | 39.7 | 34.5 | 29.6 | 29.9 | 34.1 | 43.0 | 33.8 | 29.2 | 24.6 | 24.5 | 28.8 |
| vs. GLIGEN | | **+8.1** | **+5.9** | **+2.7** | **+2.6** | **+0.4** | **+2.8** | **+9.3** | **+6.2** | **+3.7** | **+2.7** | **+0.9** | **+3.6** |
| 3DIS+MIGC | | 76.8 | 70.2 | 72.3 | 66.4 | 68.0 | 69.7 | 68.0 | 60.7 | 62.0 | 55.8 | 57.3 | 59.5 |
| vs. MIGC | | **+2.0** | **+4.0** | **+4.9** | **+1.1** | **+1.9** | **+2.6** | **+5.0** | **+6.0** | **+6.7** | **+3.4** | **+4.1** | **+4.8** |

Figure 4: **Qualitative results on the COCO-MIG (§4.3)**.

which requires training for rendering, 3DIS shows a **5**% increase in IASR, while also allowing the use of higher-quality models, such as SD2 and SDXL, to generate more visually appealing results. Importantly, the proposed 3DIS approach is not mutually exclusive with existing adapter methods. For instance, combinations like 3DIS+GLIGEN and 3DIS+MIGC outperform the use of adapter methods alone, delivering superior performance. Fig. 4 offers a visual comparison between 3DIS and other SOTA methods, where it is evident that 3DIS not only excels in scene construction but also demonstrates strong capabilities in instance detail rendering. Furthermore, 3DIS is compatible with a variety of base models, offering broader applicability compared to previous methods.

## 4.4 ABLATION STUDY

**Constructing scenes with depth maps.** Tab. 3 demonstrates that generating scenes in the form of depth maps, rather than directly producing RGB images, enables the model to focus more effectively on coarse-grained categories, structural attributes, and the overall scene composition. This approach leads to a **3.3**% improvement in AP and a **4.1**% increase in $AP_{75}$.

**Tuning of the Text-to-depth model.** Tab. 3 demonstrates that, compared to using LDM3D directly, fine-tuning LDM3D with pyramid diffusion as our base text-to-depth generation model

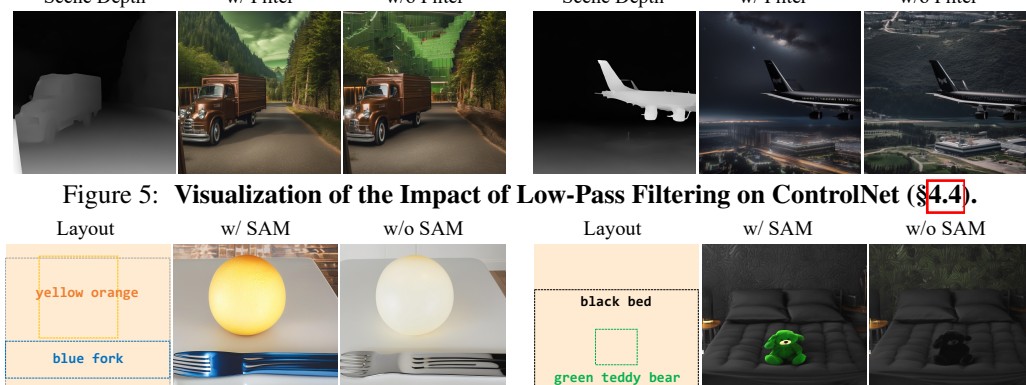

Figure 5: **Visualization of the Impact of Low-Pass Filtering on ControlNet (§4.4).**

Figure 6: **Visualization of the Impact of SAM-Enhancing Instance Location (§4.4).**

results in a **1.3**% improvement in AP and a **2.2**% increase in $AP_{75}$. These improvements stem from the fine-tuning process, which encourages the depth generation model to focus more on recovering low-frequency components, benefiting the global scene construction.

**Augmenting instance descriptions by removing adjectives.** The data presented in Tab. 3 indicate that during the training of layout-to-depth adapters, augmenting instance descriptions by removing fine-grained attribute descriptions allows the model to focus more on the structural of the instances and the overall scene construction. This approach ultimately results in a **2.8**% improvement in AP and a **3.0**% increase in $AP_{75}$.

Table 3: **Ablation study on scene generation (§4.4).**

| method | $AP/AP_{50}/AP_{75}$ ↑ | MIoU ↑ | FID ↓ |
|---|---|---|---|
| w/o using depth | 53.5 / 81.8 / 58.3 | 72.2 | 24.1 |
| w/o aug data | 54.0 / 78.4 / 59.4 | 73.3 | 23.5 |
| w/o tuning LDM3D | 55.5 / 81.9 / 60.2 | 72.8 | 25.2 |
| w/ all | **56.8 / 82.3 /62.4** | **73.3** | **23.2** |

**Low-Pass Filtering on the ControlNet.** Fig. 5 shows that filtering out high-frequency noise from ControlNet's feature maps improves the overall quality of the generated images, resulting in more accurate scene representation. Moreover, as indicated in Tab. 4, this process does not affect the Instance Attribute Success Ratio (IASR) and MIoU when rendering fine details.

Table 4: **Ablation study on rendering (§4.4).**

| method | IASR ↑ | MIOU ↑ | FID ↓ |
|---|---|---|---|
| w/o Low-Pass Filter | 55.87 | 46.93 | 24.50 |
| w/o SAM-Enhancing | 52.42 | 45.17 | 23.67 |
| w/ all | **56.01** | **47.01** | **23.24** |

**SAM-Enhancing Instance Location.** Fig. 6 illustrates that utilizing SAM for more precise instance location effectively prevents rendering conflicts caused by layout overlaps, ensuring accurate rendering of each instance's fine-grained attributes. As shown in Tab. 4, enhancing instance localization with SAM improves the Instance Attribute Success Ratio (IASR) by **3.19**% during rendering.

## 4.5 UNIVERSAL RENDERING CAPABILITIES OF 3DIS

**Rendering based on different-architecture models**. Fig. 1, 3, and 4 present the results of 3DIS rendering details using SD2 and SDXL without additional training. The results demonstrate that 3DIS not only leverages the enhanced rendering capabilities of these more advanced base models, compared to SD1.5, but also preserves the accuracy of fine-grained instance attributes.

**Rendering based on different-style models**. Fig. 7 presents the results of 3DIS rendering using various stylistic model variants (based on the SDXL architecture). As shown, 3DIS can incorporate scene depth maps to render images in diverse styles while preserving the overall structure and key instance integrity. Furthermore, across different styles, 3DIS consistently enables precise control over complex, fine-grained attributes, as illustrated by the third example in Fig. 7, where "Dotted colorful wildflowers, some are red, some are purple" are accurately represented.

**Rendering Specific Concepts**. 3DIS renders details leveraging pre-trained large models, such as SD2 and SDXL, which have been trained on extensive corpora. This capability allows users to render specific concepts. As demonstrated in Fig. 8, 3DIS precisely renders human details associated with specific concepts while preserving control over the overall scene.

| Layout | Scene Depth map | SDXL | AutismMix | Watercolor (LoRA) | Pixel Art (LoRA) |

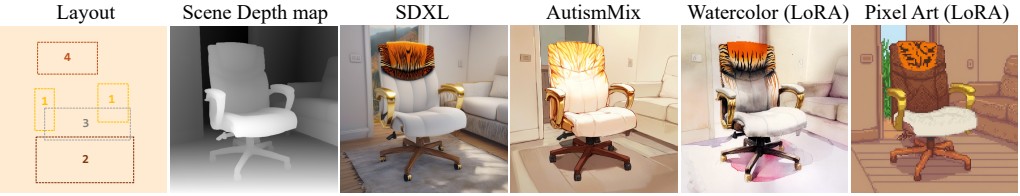

**1)** A luxurious **chair armrest** crafted from **polished gold**, with a smooth and reflective surface. **2)** **Chair legs**, carved from solid, high-quality wood, with a natural, rich grain. **3)** A **chair seat**, crafted from luxurious white velvet, soft to the touch and with a smooth, velvety finish that exudes elegance. **4)** A **chair backrest** features an exquisite, tiger-like pattern of **orange-yellow** stripes.

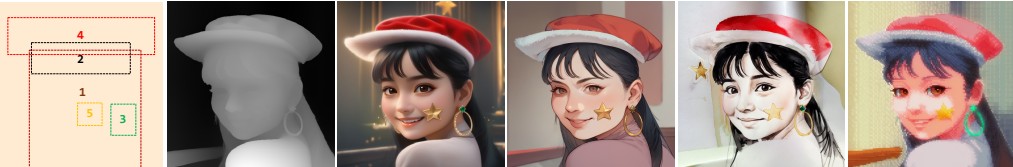

**1)** A cute **girl**, **black** hair, **brown** eyes, smiling sweetly, looking at viewer, soft expression. **2)** Glossy **black hair**, smooth and slightly wavy. **3)** A shiny jewel **earring**, embedded with **emerald** stones. **4)** A Santa-style **hat**, with a **red** body and white fluffy trim. **5)** A **Gold** star-shaped **sticker** on cheek, metallic shine.

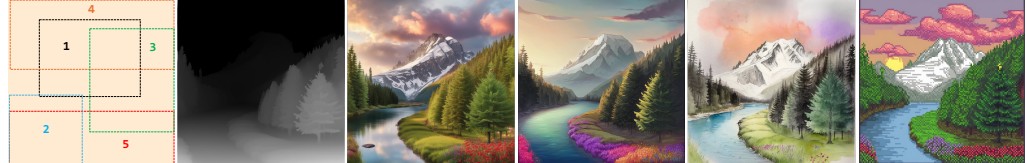

**1)** A towering snow-capped **mountain** with sprawling glacier. **2)** A crystal-clear **blue river**, gently flowing with soft ripples. **3)** A dense **forest** of **dark green** tall trees with lush foliage. **4)** Soft, warm-toned **clouds** illuminated by the colors of the sunset, blending hues of **orange**, **pink**, and **purple**. **5)** Dotted colorful **wildflowers**, some are **red**, some are **purple**.

Figure 7: **Rendering results based on different-style models (§4.5).**

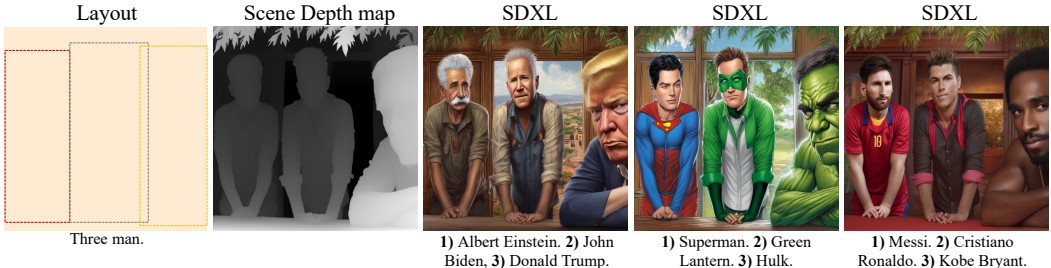

Three man.

**1)** Albert Einstein. **2)** John Biden, **3)** Donald Trump.   **1)** Superman. **2)** Green Lantern. **3)** Hulk.   **1)** Messi. **2)** Cristiano Ronaldo. **3)** Kobe Bryant.

Figure 8: **Rendering results on specific concepts (§4.5).**

## 5 CONCLUSION

We propose a novel 3DIS method that decouples image generation into two distinct phases: coarse-grained scene depth map generation and fine-grained detail rendering. In the scene depth map phase, 3DIS trains a Layout-to-Depth network that focuses solely on global scene construction and the coarse-grained attributes of instances, thus simplifying the training process. In the detail rendering phase, 3DIS leverages widely pre-trained ControlNet models to generate images based on the scene depth map, controlling the scene and ensuring that each instance is positioned accurately. Finally, our proposed detail renderer guarantees the correct rendering of each instance's details. Due to the training-free nature of the detail rendering phase, our 3DIS framework utilizes the generative priors of various foundational models for precise rendering. Experiments on the COCO-Position benchmark demonstrate that the scene depth maps generated by 3DIS create superior scenes, accurately placing each instance in its designated location. Additionally, results from the COCO-MIG benchmark show that 3DIS significantly outperforms previous training-free rendering methods and rivals state-of-the-art adapter-based approaches. We envision that 3DIS will enable users to apply a wider range of foundational models for multi-instance generation and be extended to more applications. In the future, we will continue to explore the integration of 3DIS with DIT-based foundational models.

**Acknowledgements.** This work was supported in part by the Natural Science Foundation of Zhejiang Province (LDT23F02023F02) and Fundamental Research Funds for the Zhejiang Provincial Universities (226-2024-00208).

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
