# Appendix

## A  INFERENCE EFFICIENCY ANALYSIS

**Inference Efficiency Analysis of 3DIS.** The 3DIS framework generates high-resolution images in three sequential stages: **1) The Layout-to-Depth Model**, which creates a coarse-grained scene depth map; **2) The Segmentation Model**, which extracts the precise shape of each instance from the scene depth map; **3) The Detail Renderer**, which uses various foundational models (SD2, SDXL, etc.) to produce the final high-resolution image. We evaluated the inference efficiency of these stages using an NVIDIA A100 GPU. Our test involved a layout with 10 instances, and we assessed the inference time for each stage over 50 runs to calculate an average time:

- **Layout-to-Depth Model:** Given that the global scene depth map does not require high granularity, the UniPCMultistepScheduler (Zhao et al., 2023) is employed for only 30 steps. The average time to generate a depth map is **5.66** seconds.
- **Segmentation Model:** We utilize the SAM model to segment the generated scene depth maps and get refined layouts. The refinement process by SAM takes **0.14** seconds.
- **Detail Renderer:** We use the EulerDiscreteScheduler (Karras et al., 2022) for 50 steps. The time for the SD1.5 model to render a $512 \times 512$ image is **5.27** seconds, the time for the SD2 model to render a $768 \times 768$ image is **11.28** seconds, and the time for the SDXL model to render a $1024 \times 1024$ image is **22.75** seconds.

Table A: **Average inference time of different layout-to-Image model.**

|  | GLIGEN | InstanceDiff | MIGC | 3DIS (SD1.5) | 3DIS (SD2) | 3DIS (SDXL) |
|---|---|---|---|---|---|---|
| **Inference Time (s)** | 12.75 | 42.48 | 6.81 | 11.07 | 17.08 | 28.55 |
| **Resolution** | 512 | 512 | 512 | 512 | 768 | 1024 |

**Inference Efficiency Comparison.** We conducted comparative experiments to evaluate the performance of various state-of-the-art (SOTA) methods, including GLIGEN (Li et al., 2023b), Instance Diffusion (Wang et al., 2024), and MIGC (Zhou et al., 2024), using NVIDIA A100 GPU. All models were tested using the default configurations in their GitHub repositories. We evaluated the inference efficiency of these stages using an NVIDIA A100 GPU. Our test involved a layout with 10 instances, and we assessed the inference time for each stage over 50 runs to calculate an average time. The experimental results are shown in Tab. A. The conclusions are as follows:

- **3DIS demonstrates faster inference speeds with SD1.5.** Since the scene depth map generated by 3DIS does not require too high granularity, the speed of generating the scene depth map is very fast. The average inference time of 3DIS + SD1.5 is 11.07s, even faster than GLIGEN and Instance Diffusion, which are based on the same SD1.5 base model.
- **3DIS demonstrates acceptable inference speeds with SD2 and SDXL.** As we increase model capacity and image resolution, the inference time for 3DIS also rises. Rendering times are **17.08** seconds for SD2 and **28.55** seconds for SDXL, which we consider to be acceptable. Additionally, our experiments show that using 3DIS with SDXL even achieves faster processing speeds than InstanceDiffusion. As discussed in Section 4.3, the performance of 3DIS + SDXL on COCO-MIG slightly surpasses that of InstanceDiffusion, demonstrating the practicality and efficiency of our 3DIS framework comprehensively.

## B  RESULTS OF OVERLAPPING LAYOUTS WITH DEPTH AMBIGUITY

**3DIS allows for direct adjustment of the instance font-back according to user specifications (see Fig. A).** Although our layout-to-depth model does not explicitly incorporate instance front-back ordering during the training process or network design, we found that certain training-free methods can still achieve control over instance front-back ordering. Specifically, our layout-to-depth model integrates layout information via a layout adapter (i.e., MIGC). For N instances, this adapter

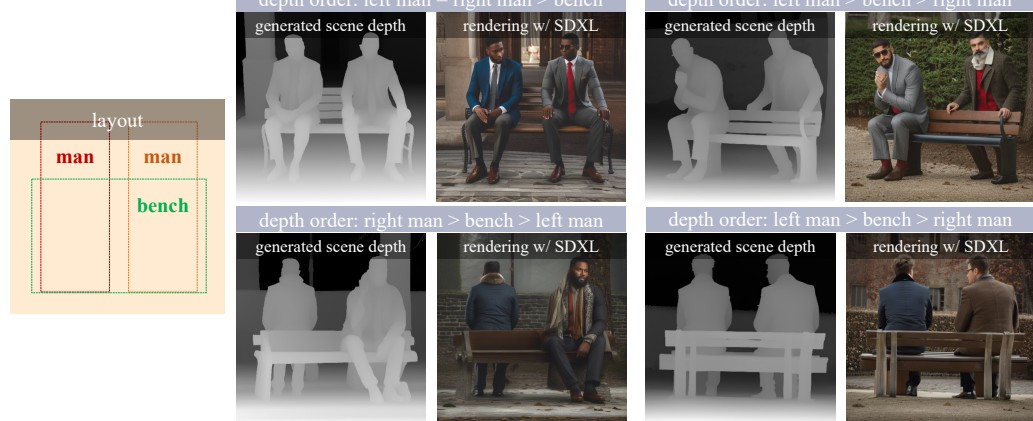

Figure A: **User-specified Front-Back Instance Ordering in Scene Depth Map Generation (§B).** For layouts with depth ambiguity, 3DIS allows for direct adjustment of the instance ordering according to user specifications, generating distinct scene depth maps and rendering them accordingly.

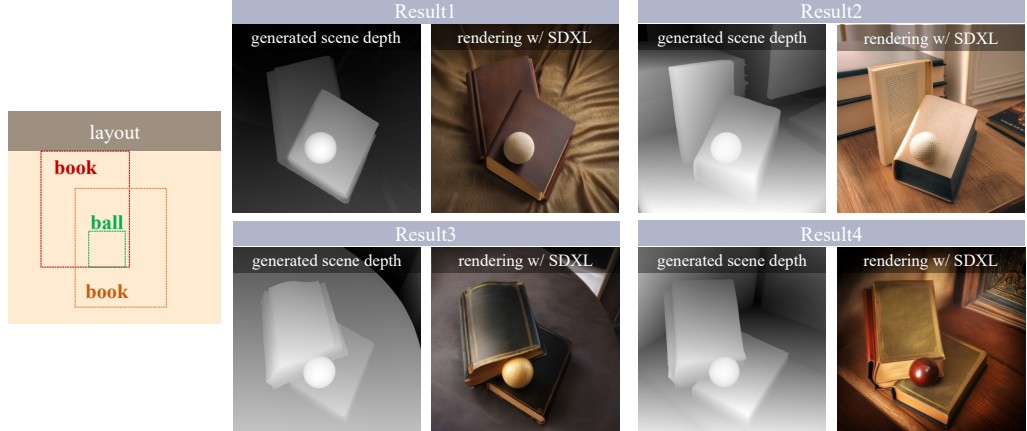

Figure B: **Automatic Front-Back Instance Ordering in Scene Depth Map Generation. (§B).** For the same overlapping layout with depth ambiguity, 3DIS can generate different scene depth maps with varying seeds, ensuring that the generated scenes adhere to the specified layout. Instances overlapping in the layout may display varying front-back order across different generated outcomes.

encodes them into N tokens, which are then injected into image features through a newly trainable Cross-Attention layer. For each specific pixel in the image features, the Cross-Attention layer uses a softmax function to determine the scale score of each instance token. Notably, we discovered that by adjusting the scale score (before the softmax function) of a token, we can control the relative depth ordering of instances (e.g., larger scale scores bring instances to the foreground, while smaller scale scores push them to the background). By adjusting the scale scores for each instance, we can thus control the front-back ordering within overlapping regions of the scene.

**3DIS is capable of automatically adjusting the depth order of instances without explicit specifications (see Fig. B).** As illustrated in Fig. B, the overlap of instances can be categorized into two types: 1) Complete overlap, as seen in the relationship between the ball and the books. As the ball's bounding box is fully enclosed within the books' bounding boxes, 3DIS typically generates it in the foreground to prevent it from disappearing. 2) Partial overlap, as in the case of the two books. In this scenario, depending on the seed, the front-back ordering of the books may vary, resulting in different depth placements across the generated scenes.

## C  COMPARISON OF LDM3D AND RICHDREAMER

Upon investigation, we identified RichDreamer (Qiu et al., 2024) and LDM3D (Stan et al., 2023) as the primary models employed for text-to-depth generation. To compare their performance, we

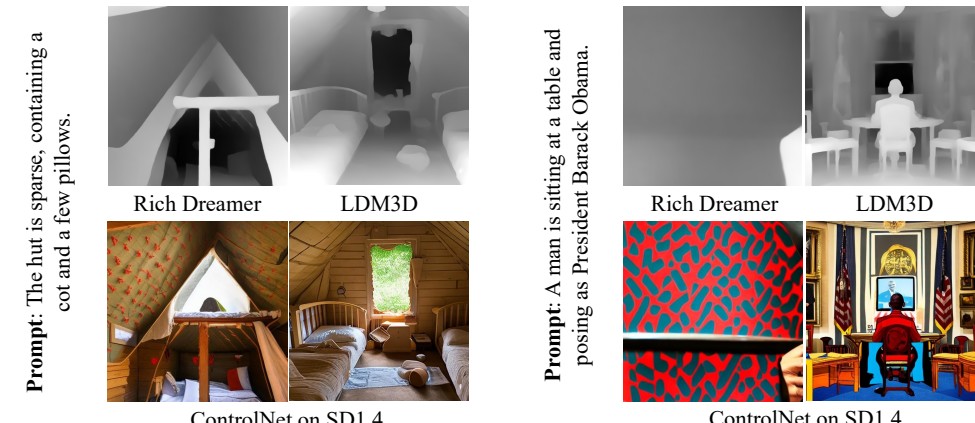

Figure C: **Comparison of LDM3D and RichDreamer.**

utilized prompts from the COCO2014 dataset as input for both models, with the corresponding results illustrated in Fig. C. Our analysis indicates that LDM3D demonstrates a superior ability to preserve the original SD1.4 priors, resulting in enhanced text comprehension and more precise control over scene generation. In contrast, RichDreamer exhibits certain shortcomings: (i) it often misses semantic details or omits entire objects in the depth maps, as seen in cases where essential elements like the **cot** and **man** are entirely absent; (ii) the depth maps produced by RichDreamer frequently suffer from artifacts such as blotches or thread-like distortions, particularly when used in conjunction with ControlNet. Therefore, after a thorough comparison, we selected LDM3D as the base model for text-to-depth generation in our 3DIS system.

## D    VISUALIZATION ON THE IMPACT OF THE LDM3D FINE-TUNING

Although LDM3D is capable of generating relatively good depth maps, several issues remain: (i) Since LDM3D was trained using depth maps extracted from the DPT-Large Model (Ranftl et al., 2021) , the resulting image quality is relatively poor. (ii) As a diffusion model trained by Gaussian noise, LDM3D exhibits limited ability to recover low-frequency content (Guttenberg, 2023). This is clearly illustrated in Fig. D, where the generated depth maps struggle to produce large uniform color blocks. Moreover, the average color value of the depth maps tends to converge towards the initial noise, whose mean value is close to 0. This constraint places a harmful limitation on text-to-depth generation.

To address (i), we fine-tuned the model using depth maps extracted from the latest Depth-Anything V2 model. For (ii), we adopted pyramid noise instead of Gaussian noise, which helps mitigate the constraints on text-to-depth generation. As shown in Fig. D, the fine-tuned LDM3D model is capable of generating depth maps with higher contrast and improved overall quality.

## E    EXAMPLES OF GENERATED ANNOTATION

**Text-depth pair in LAION-art (see Fig. E).** The text-to-depth pair is essential for training our text-to-depth model. To obtain high-quality RGB images, we selected images from LAION-art with an aesthetic score greater than 8.0 and a resolution exceeding 512. Given that the text descriptions in LAION-art are often noisy, we chose to use the BLIP2 (Li et al., 2023a) model to generate more accurate captions. As shown in Fig. E, BLIP-generated captions can precisely capture the key information of the image. While the model still has limitations in describing certain fine-grained attributes—such as the color in the first example of the second row, where the description is inaccurate—this is not crucial for depth map generation, where fine-grained details are less significant. We use the Depth Anything V2 model to obtain high-quality depth maps corresponding to each image, which, together with the generated captions, form the text-depth pairs for training.

**Layouts in COCO dataset (see Fig. F).** The COCO (Lin et al., 2015) dataset contains images along with corresponding human-annotated natural language descriptions. For example, in the first image

of the first row of Fig. F, the annotated description is: "A white vase filled with a mix of white and pink flowers on a porch railing." To further extract descriptions for each instance, we use the Stanza (Qi et al., 2020) parser to analyze the noun phrases in the sentence, such as "A white vase," "A mix of white and pink flowers," and "porch railing." Based on these instance descriptions, we employ Grounding-DINO (Liu et al., 2023) to detect the bounding boxes of each instance, thereby obtaining the layout of the entire image and detailed descriptions of the instances.

## F   USER STUDY

We conducted a user study to evaluate user preferences, selecting three methods for comparison: 3DIS, MIGC (Zhou et al., 2024), and InstanceDiffusion (Wang et al., 2024). For each participant in the user study, we randomly selected 30 images from the COCO-MIG benchmark and asked them to rank the images based on their preference. A total of 30 participants were invited, and the aggregated results are presented in Fig. G. The results indicate that, compared to MIGC and InstanceDiffusion, 3DIS was generally preferred by users. This preference is attributed partly to 3DIS's superior control over spatial positioning and also to its ability to leverage stronger foundational models for rendering in a training-free manner, resulting in higher-quality images.

## G   ADDITIONAL EXAMPLES OF 3DIS

**Additional examples of controlling shape and pose (see Fig. H).** Under the same layout, 3DIS can generate different scene depth maps and control coarse-grained attributes of different instances, such as shape and pose. As shown in Fig. H(a), we can freely change the shape of the cake and table within the same layout. Similarly, in Fig. H(b), we can adjust each person's pose.

**Additional examples of complicated layouts (see Fig. I).** For highly complex layouts, 3DIS reliably ensures accurate generation results. In Fig. I(a), 3DIS successfully creates a counterfactual scene where an ice mountain, volcano, mallard, swallow, and cherry coexist harmoniously. In Fig. I(b), 3DIS precisely renders each part of an eagle according to the specified input.

**Additional examples of COCO-position benchmark.**  Fig. J presents additional results of scene depth map generation using our 3DIS system. The results demonstrate that, even with complex layouts, 3DIS effectively understands and generates cohesive scenes, harmoniously placing all objects within them. Furthermore, even in cases of significant overlap, such as the five suitcases in the fifth row, 3DIS handles the arrangement with precision, maintaining clear object separation and preventing blending.

**Additional examples of COCO-MIG benchmark.** Fig. L presents additional results of 3DIS on the COCO-MIG dataset, revealing several key advantages over the previous state-of-the-art model, MIGC. *1) 3DIS demonstrates superior scene construction capabilities*, as seen in the first and second rows, where it constructs more coherent scenes that appropriately place all specified instances—such

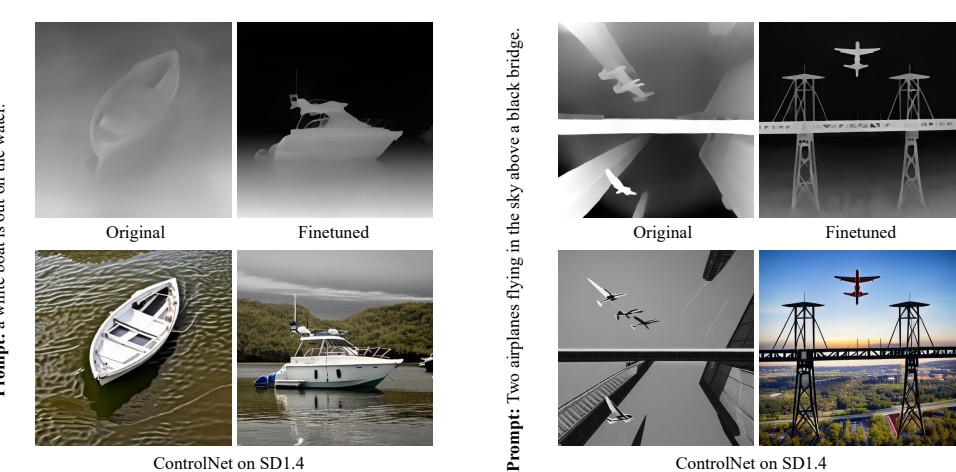

Figure D: **Comparison of original LDM3D and finetuned LDM3D.**

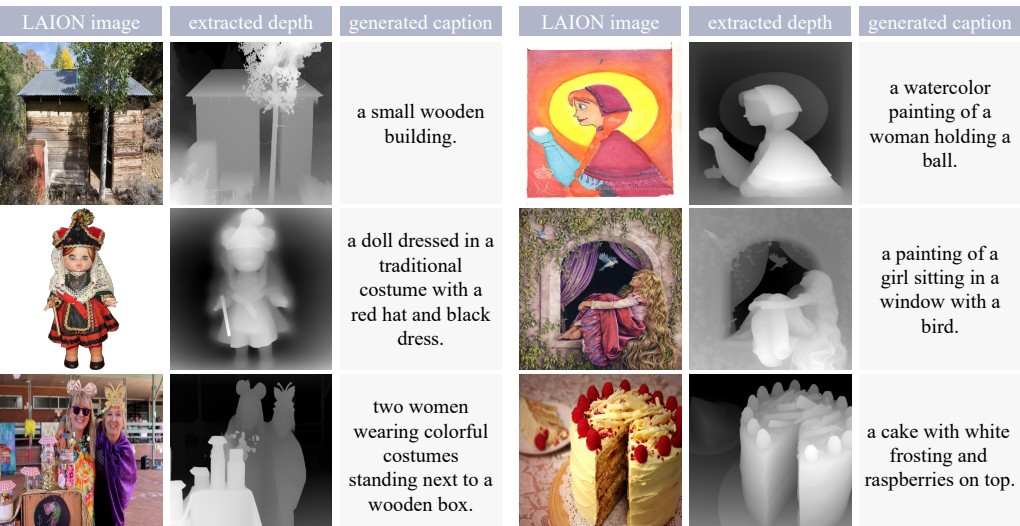

Figure E: **Examples of the generated annotation in the LAION-art dataset.** By utilizing the Depth Anything V2 model to extract depth maps and employing the BLIP2 model to generate captions corresponding to images, we can obtain high-quality text-depth pairs. These pairs will be used to train our text-to-depth model.

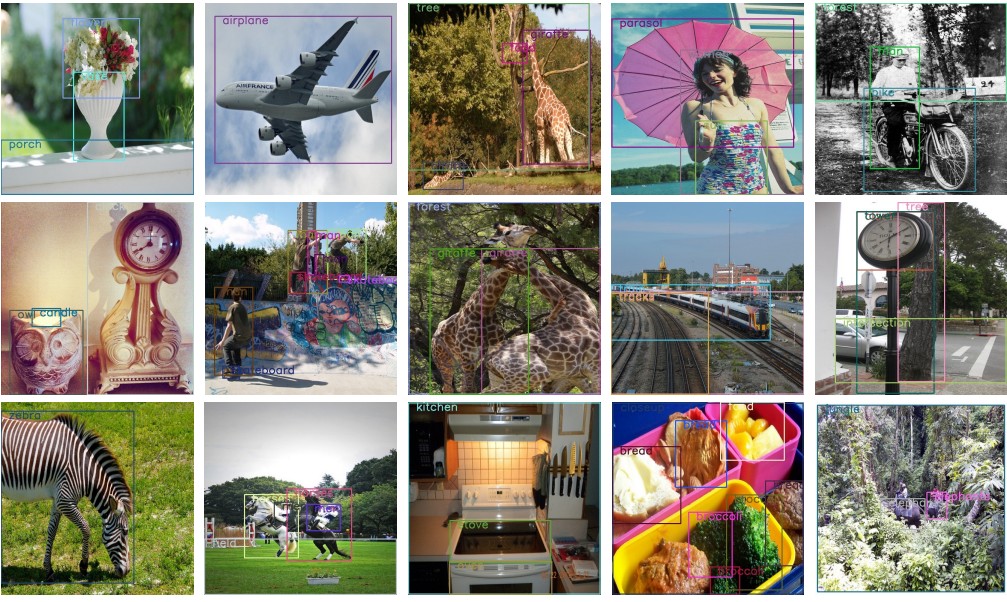

Figure F: **Examples of the generated layouts in the COCO dataset.** We have omitted the adjectives from each instance to better highlight the generated layout.

as rendering an indoor environment when prompted with "refrigerator." *2) 3DIS exhibits enhanced detail rendering*, as shown in the fourth to sixth rows. By leveraging the more advanced SDXL model in a training-free manner, 3DIS outperforms MIGC, which primarily relies on SD1.5, producing more visually appealing and structurally refined results. *3) 3DIS handles smaller instances better*, as demonstrated in the third row with the "red bird" and "yellow dog." Its ability to render at higher resolutions using SDXL leads to clearer and more accurate depictions of these smaller objects. **Finally, 3DIS excels in managing overlapping objects**, as illustrated in the seventh row, where it avoids object merging while generating the scene's depth map.

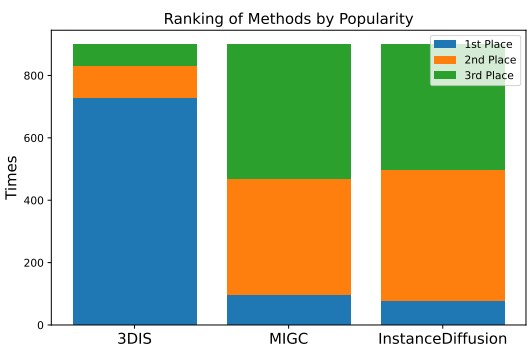

Figure G: **User Study.** Compared with the previous state-of-the-art methods, 3DIS is more popular.

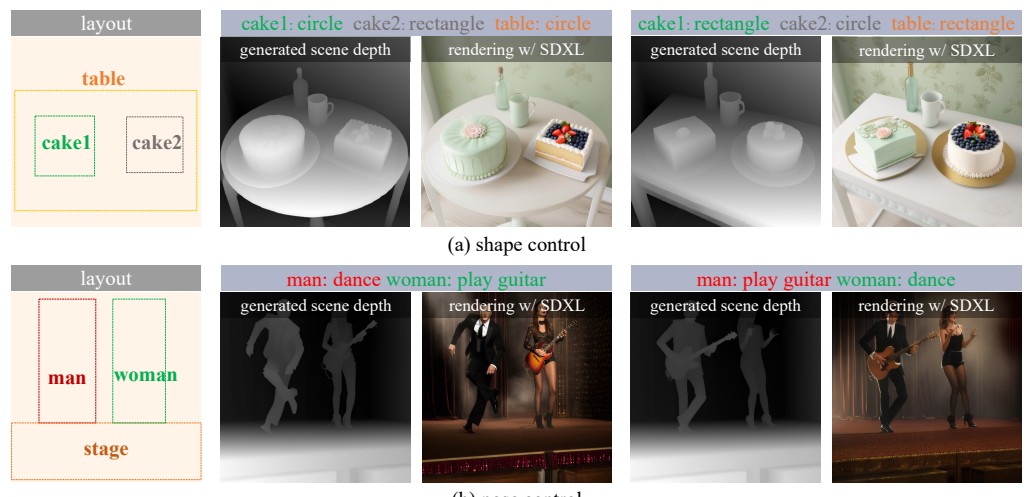

(a) shape control

(b) pose control

Figure H: **Additional Generated Examples.** With the same layout, 3DIS can modify the shape and pose of each instance automatically.

## H   MORE DETAILS OF THE INFERENCE PIPELINE

**Scene Depth Maps Generation.** Given that the scene depth map primarily focuses on coarse-grained attributes for scene construction and instance placement, it is unnecessary to generate extensive detail at this stage. Therefore, unlike previous methods (Zhou et al., 2024; Li et al., 2023b), which typically employ 50 steps for scene generation, we use only 30 steps, utilizing the UniPCMultistepScheduler (Zhao et al., 2023). Additionally, the Classifier-Free Guidance (Ho, 2022) (CFG) scale is set to 7.5.

**Detail Rendering.** In this phase, we utilize the EulerDiscreteScheduler (Karras et al., 2022) for 50 steps to render details meticulously. To reduce high-frequency noise in the generated depth map and to emphasize low-frequency scene information, we apply an FFT filter to the ControlNet signals. This filtering is specifically targeted at the mid and lower resolution upper layers. Initially, we perform a Fast Fourier Transform (FFT) to centralize the zero-frequency component within the spectrum. Subsequently, we design and implement a frequency mask that attenuates high frequencies beyond the central region extending to H/4 and W/4 from the center, setting a scale of 0.5 to predominantly preserve the central region, where H and W represent the height and width of the residual features injected from the ControlNet. An inverse FFT is then conducted to transform the data back to the spatial domain. The outcome is a refined version of the ControlNet feature, enriched with primarily low-frequency scene information.

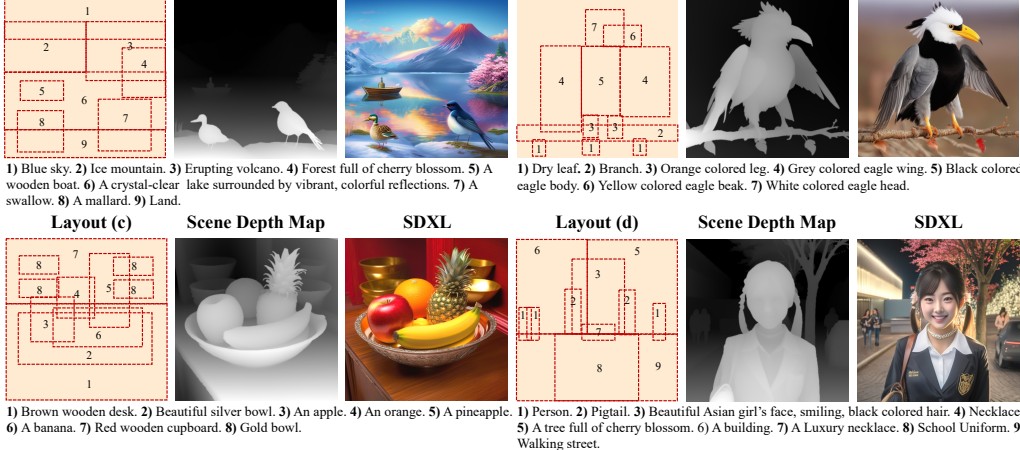

1) Blue sky. 2) Ice mountain. 3) Erupting volcano. 4) Forest full of cherry blossom. 5) A wooden boat. 6) A crystal-clear lake surrounded by vibrant, colorful reflections. 7) A swallow. 8) A mallard. 9) Land.

1) Dry leaf. 2) Branch. 3) Orange colored leg. 4) Grey colored eagle wing. 5) Black colored eagle body. 6) Yellow colored eagle beak. 7) White colored eagle head.

1) Brown wooden desk. 2) Beautiful silver bowl. 3) An apple. 4) An orange. 5) A pineapple. 6) A banana. 7) Red wooden cupboard. 8) Gold bowl.

1) Person. 2) Pigtail. 3) Beautiful Asian girl's face, smiling, black colored hair. 4) Necklace. 5) A tree full of cherry blossom. 6) A building. 7) A Luxury necklace. 8) School Uniform. 9) Walking street.

Figure I: **Additional Generated Examples.** 3DIS also demonstrates robust generation capabilities for complex layouts.

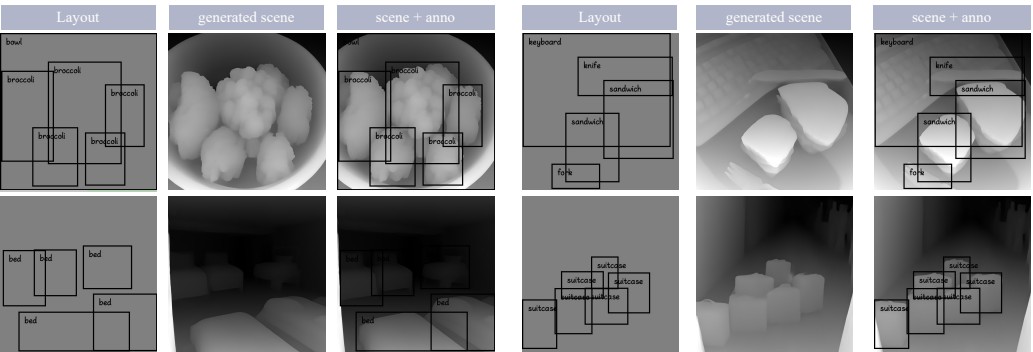

Figure J: **More results of the generated scene depth map.**

# I  LIMITATION

Although 3DIS leverages various foundation models for rendering fine instance details, its scene construction continues to rely on the less advanced SD1.5 model. This dependency limits 3DIS's capacity to accurately generate complex structures, particularly in tasks that SD1.5 struggles with, such as text rendering, intricate shapes, or highly detailed spatial configurations. For example, if we aim to generate a high-quality strawberry cake with the text "ICLR" written on it, 3DIS is unlikely to generate scene depth maps correctly (e.g., the wrong "L" letter in Fig. K). Addressing this limitation in future work could involve the development of specialized datasets aimed at enhancing the model's proficiency in handling complex structures, such as MARIO-10M (Chen et al., 2023), thereby improving the overall robustness and versatility of 3DIS in a broader range of applications.

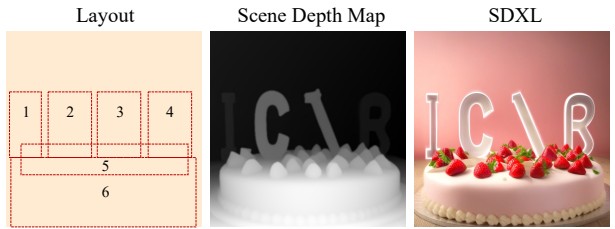

1) Letter "I". 2) Letter "C". 3) Letter "L". 4) Letter "R". 5) A huge delicious cake. 6) Many red strawberries.

Figure K: **Failure case of the 3DIS.**

## J    EMPHASIZING THE CONTRIBUTION OF 3DIS

**Motivation:** Previous layout-adapter methods have only released weights for SD1.5, necessitating retraining for deployment on more powerful models like SD2 and SDXL, which is both time-consuming and burdensome. Our 3DIS method divides MIG into two parts: scene depth construction and detail rendering. For scene depth construction, we only train the layout adapter once for scene depth map generation, focusing primarily on coarse-grained semantics, which is adequately handled by the SD1.5 model. For detail rendering, 3DIS employs various stronger models and their widely pre-trained ControlNet in a training-free manner, allowing users to benefit from the enhanced performance of increasingly powerful models.

**Technology:** Our 3DIS method restructures Multi-Instance Generation into two phases: constructing a scene depth map and training-free detail rendering. This process differs significantly from previous approaches and has two notable features: 1) Generating a scene depth map rather than an RGB image in the first stage allows the layout adapter to focus on coarse-grained attributes, effectively improving its spatial control capabilities and handling overlapping scenarios with added depth knowledge. 2) The training-free detail rendering method enables users to utilize various foundational models and their widely available pre-trained ControlNet for rendering details directly.

**Experiment Results:** Our experiments show that our method surpasses previous approaches in location control and allows the use of various foundation models for rendering without additional training costs, resulting in markedly superior outcomes in detail rendering.

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

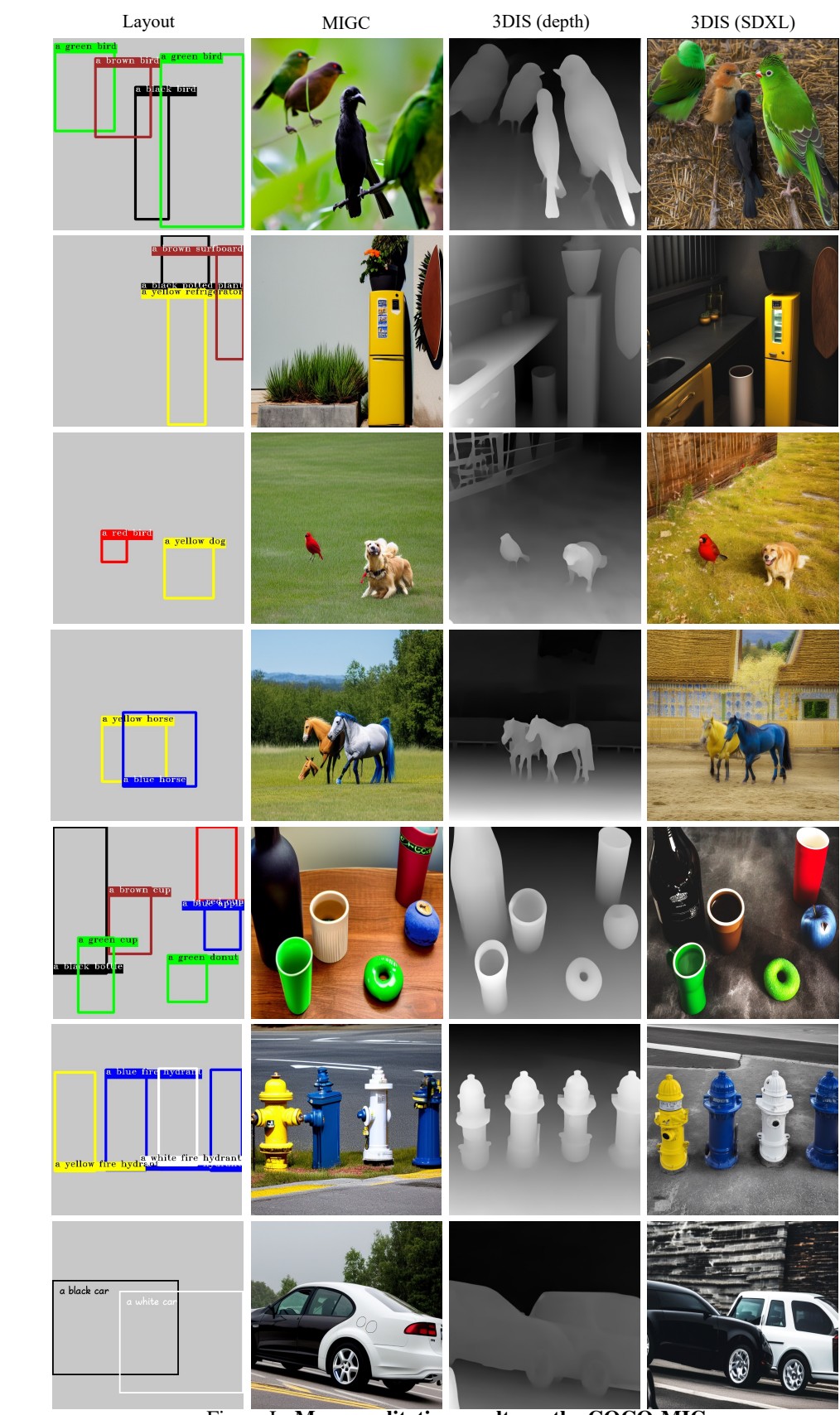

Figure L: **More qualitative results on the COCO-MIG.**