# OpenReview forum: "3DIS: Depth-Driven Decoupled Image Synthesis for Universal Multi-Instance Generation"
_ICLR.cc/2025/Conference — ICLR 2025 Spotlight_

### Official Review · Reviewer_dKuo · 2024-10-30

**Soundness:** 3
**Presentation:** 3
**Contribution:** 3
**Rating:** 8
**Confidence:** 4

**Summary:**

In this paper, the authors propose a novel multi-instance generation method, which uses a depth map as the intermediate generation step to decouple the whole process into two stages. Based on the depth map, the following rendering process only applies existing models, such as SAM, ControlNet, SD, and so on. Experiments show that the generated image is more in line with the given layout and description.

**Strengths:**

1. The proposed method is flexible and can combine many existing generation models.
2. The location improvement is pretty obvious compared to previous SoTA methods.

**Weaknesses:**

1. The locations of the cat and lamp are not correct in Figure 1.
2. The details of the comparison should be clarified. For example, it mentions that MIGC only supports SD1.5 in Figure 1, but the visual results of the proposed method in Figures 3 and 4 are based on SD2 and SDXL. In addition, it's unclear which version of SD is used in Table 1.
3. It's better to provide some inference efficiency analysis since the proposed method contains multiple steps.
4. The visual quality is not very satisfactory. It's better to apply user study to evaluate user preferences.
5. It's better to add the image quality metrics to the ablation study in Tables 3 and 4.

**Questions:**

Please see the weakness.

---

> ### Author Response · Authors · 2024-11-19
>
> We sincerely **appreciate your recognition of the good performance and strong flexibility of our 3DIS approach**. Thank you very much for your valuable comments!
>
> We will address all the concerns point by point.
>
>
>
> **Weakness 1**: The locations of the cat and lamp are not correct in Figure 1.
>
> **Answer**: We appreciate your observation regarding the incorrect positions of the cat and lamp in Figure 1. The positions of the cat and lamp in Figure 1 **have been corrected**, and **we have thoroughly reviewed and verified the accuracy of all the figures to ensure they correctly represent the intended conten**t.
>
> ---
>
> **Weakness 2.1**: The details of the comparison should be clarified. For example, it mentions that that MIGC only supports SD1.5 in Figure 1, but the visual results of the proposed method in Figures 3 and 4 are based on SD2 and SDXL.
>
> **Answer**: Thank you for your feedback regarding the details of our comparison. In Figures 3 and 4, we **focused on demonstrating the core functionality of 3DIS**, which is its ability to **train-freely integrate stronger models like SD2 and SDXL** to generate higher-quality images. In contrast, **previous adapter methods required retraining an adapter for each base model, and the community has only released weights for these adapters on SD1.5**, thus limiting their rendering capabilities to SD1.5.
>
> ---
>
> **Weakness 2.2**: In addition, it's unclear which version of SD is used in Table 1.
>
> **Answer**: Thank you for your thorough consideration! We agree that this is an important detail to clarify. **The results of 3DIS in Table 1 are based on 3DIS + SD1.5**, as the COCO-POS benchmark is primarily designed to evaluate the model's ability to control the position of objects without focusing on fine-grained details such as color. **We have added this explanation to the main text for clarity.**

---

> ### Author Response · Authors · 2024-11-19
>
> **Weakness 3**: It's better to provide some inference efficiency analysis since the proposed method contains multiple steps.
>
> **Answer**: We greatly appreciate your valuable feedback! Now we have added an inference efficiency analysis in the **Appendix's** **Section A**.
>
> **Firstly, We conducted an efficiency analysis of 3DIS.** The 3DIS framework generates high-resolution images in three sequential stages: **1)The Layout-to-Depth Model**, which creates a coarse-grained scene depth map; **2) The Segmentation Model**, which extracts the precise shape of each instance from the scene depth map; **3) The Detail Renderer**, which uses various foundational models (SD2, SDXL, etc.) to produce the final high-resolution image. We evaluated the inference efficiency of these stages using an NVIDIA A100 GPU. Our test involved a layout with 10 instances, and we assessed the inference time for each stage over 50 runs to calculate an average time:
>
> + **Layout-to-Depth Model**: The average time to generate a depth map is **5.66 seconds.**
> + **Segmentation Model**: We utilize the SAM model to segment the generated scene depth maps and get refined layouts. The refinement process by SAM takes **0.14 seconds.**
> + **Detail Renderer**: We use the EulerDiscreteScheduler for 50 steps.
>   - The time for the SD1.5 model to render a 512 x 512 image is **5.27 seconds**,
>   - the time for the SD2 model to render a 768 x 768 image is **11.28 seconds**,
>   - and the time for the SDXL model to render a 1024 x 1024 image is **22.75 seconds**.
>
> **Secondly, We also conducted a comparison to test other adapter rendering methods**, including GLIGEN (Li et al., 2023b), Instance Diffusion (Wang et al., 2024), and MIGC (Zhou et al., 2024), using their default configurations on NVIDIA A100 GPUs. The results are summarized in the Table below:
>
> | Model                  | GLIGEN | InstanceDiff | MIGC | 3DIS (SD1.5) | 3DIS (SD2) | 3DIS (SDXL) |
> | ---------------------- | ------ | ------------ | ---- | ------------ | ---------- | ----------- |
> | **Inference Time (s)** | 12.75  | 42.48        | 6.81 | 11.07        | 17.08      | 28.55       |
> | **Resolution**         | 512    | 512          | 512  | 512          | 768        | 1024        |
>
>
> We can arrive at the following conclusion:
>
> + **3DIS demonstrates faster inference speeds with SD1.5**. Since the scene depth map generated by 3DIS does not require too high granularity, the speed of generating the scene depth map is very fast. The average inference time of **3DIS + SD1.5 is 11.07s**, even faster than GLIGEN and Instance Diffusion, which are based on the same SD1.5 base model.
> + **3DIS demonstrates acceptable inference speeds with SD2 and SDXL.** As we increase model capacity and image resolution, the inference time for 3DIS also rises. Rendering times are **17.08 seconds for SD2 and 28.55 seconds for SDXL, which we consider to be acceptable**. Additionally, our experiments show that using 3DIS with SDXL **even achieves faster processing speeds than InstanceDiffusion.** As discussed in Section 4.3, the performance of 3DIS + SDXL on COCO-MIG slightly surpasses that of InstanceDiffusion, demonstrating the practicality and efficiency of our 3DIS framework comprehensively.

---

> ### Author Response · Authors · 2024-11-19
>
> **Weakness 4.1**:  The visual quality is not very satisfactory.
>
> **Answer**: Thank you for your feedback regarding the visual quality of the images presented in our paper. We think the possible reasons are as follows:
>
> + Firstly, **due to space limitations, most of the demo images are resized to the same size as the SD1.5 generated images**, which may result in some **loss of detail** and make it difficult to see the advantages over SD1.5.
> + Secondly, for a fairer comparison, many of our experimental results, such as Figure 3&4&5&6, are **directly taken from the results of the COCO-POS/COCO-MIG benchmark**, **without any prompt optimization**, and **we** **did not use the SDXL refiner for post-processing**, which may also lead to some visual differences.
>
> **We carried out a user study** where we asked users about their preferences for results generated by different models, as shown in Figure G in the updated appendix. The results show that **users liked the results from the 3DIS model better than the others**. Also, the visual comparison charts in the initial paper, **Figures 3 and 4, clearly show that our method produces visually better results than previous methods.** Furthermore, we have included **additional results generated by the 3DIS model in the revised appendix** for further examination.
>
> ---
>
> **Weakness 4.2**: It's better to apply user study to evaluate user preferences.
>
> **Answer:** Thank you for your suggestion! According to your suggestion, **we have also added a user study in Appendix Section F**. The experimental results show that the vast majority of **users believe the image quality of 3DIS is better than that of previous SOTA models.** If you can provide more details about the visual quality that you are not satisfied with, we would be more than grateful and welcome further discussion with you.
>
> ---
>
> **Weakness 5:** It's better to add the image quality metrics to the ablation study in Tables 3 and 4.
>
> **Answer**: Thank you for your valuable suggestions. **We concur with your perspective and have incorporated the image quality metric, specifically the FID, into our ablation study**:
>
> **The results of the revised Table 3 are as follows:**
>
> | method           | AP/AP50/AP75   | MIoU ↑ | FID ↓ |
> | ---------------- | -------------- | ------ | ----- |
> | w/o using depth  | 53.5/81.8/58.3 | 72.2   | 24.1  |
> | w/o aug data     | 54.0/78.4/59.4 | 73.3   | 23.5  |
> | w/o tuning LDM3D | 55.5/81.9/60.2 | 72.8   | 25.2  |
> | w/ all           | 56.8/82.3/62.4 | 73.3   | 23.2  |
>
>
> **The results of the revised Table 4 are as follows:**
>
> | method              | IASR ↑ | MIoU ↑ | FID ↓ |
> | ------------------- | ------ | ------ | ----- |
> | w/o Low-Pass Filter | 55.87  | 46.93  | 24.50 |
> | w/o SAM-Enhancing   | 52.42  | 45.17  | 23.67 |
> | w/ all              | 56.01  | 47.01  | 23.24 |
>
>
> **From the results presented above, we can draw the following conclusions:**
>
> + **Tuning LDM3D with pyramid noise significantly enhances image quality.** As detailed in the appendix, fine-tuning the LDM3D with pyramid noise enables the model to focus more on restoring low-frequency information, which helps in constructing higher-quality scene depth maps, ultimately leading to improved image quality.
> + **Low-Pass Filtering substantially improves image quality.** Generated depth maps may introduce some artifacts; attenuating the high-frequency noise in the Controlnet signal through Low-Pass Filtering can mitigate the effects of these artifacts, enabling ControlNet to focus more on the overall scene construction. This results in more aesthetically pleasing images, as illustrated in Figure 5 of the initial paper.

---

> > ### Comment · Reviewer_dKuo · 2024-11-21
> > **Response**
> >
> > Thanks for the detailed responses from the authors. I think my main concerns have been addressed, so I will increase my final score to 8.

---

> > > ### Author Response · Authors · 2024-11-21
> > >
> > > Thank you for your prompt response and positive feedback on our revisions. Please let us know if you have any further questions or suggestions.
> > >
> > > We also appreciate your valuable contributions throughout the review process!

---

### Official Review · Reviewer_Pjgj · 2024-11-03

**Soundness:** 2
**Presentation:** 3
**Contribution:** 2
**Rating:** 6
**Confidence:** 3

**Summary:**

This paper introduces 3DIS for multiple instance generation(MIG) tasks in text-to-image generation. 3DIS decouples the image generation process into two stages. First, 3DIS generates a coarse scene depth map that accurately positions instances and aligns their coarse attributes. Second, it renders fine-grained instance attributes using pre-trained ControlNet on any foundation diffusion model without additional training. This approach enables seamless integration with diverse foundational models with a Depth-Controlled method, providing a robust and adaptable solution for advanced multi-instance generation.

**Strengths:**

1. Proposed a novel approach for MIG task with Depth control, the proposed method shows good generalizability and performance on different MIG benchmarks.

2. The proposed 3DIS can be seamlessly integrated into various pre-trained diffusion models, without requiring fine-tuning processes for each pretrained diffusion model.

3. The incorporation of 3D depth information can provide a better understanding of instances' attributes, thus benefiting the generation process of 2D images.

**Weaknesses:**

1. This pipeline is largely dependent on the fine-tuned Layout-to-Depth generation model, and the final output uses the foundation diffusion model to render instances in a strictly confined region(depth). There might be inconsistencies between the layout-to-depth generation model and the renderer model, thus hindering the final performance.

2. The slightest artifacts in the depth map might cause great changes in the final generation results, like unwanted additional objects or background depth.

**Questions:**

1. Can the layout-to-depth model generate diverse depth maps for the same layout? Like changing the pose, shape, and other attributes of instances?

2. It seems that sometimes the background scene can be blurry or unpleasant. Does that have something to do with the instance-wise rendering step?

3. Are there any failure cases that can make analyses on the Layout-to-depth model?

4. How long will it take to generate a single image?

---

> ### Author Response · Authors · 2024-11-19
>
> We sincerely **appreciate your recognition of the novelty, strong flexibility, and good performance of our 3DIS approach, as well as your acknowledgment of our writing**. Thank you very much for your valuable comments!
>
> We will address all the concerns point by point.
>
> **Question 1**: Can the layout-to-depth model generate diverse depth maps for the same layout? Like changing the pose, shape, and other attributes of instances?
>
> **Answer**: Thank you for the insightful question. Our layout-to-depth model can generate **diverse scene depth maps** while **adhering to the given layout**. For detailed results, please refer to the updated **Appendix's Section B and Figure B.**
>
> Moreover, Our layout-to-depth model allows for coarse-grained control over attributes such as **object shapes** and **human poses**. This can be achieved by providing specific prompts in the input layout. For detailed results, please refer to updated **Appendix's Section G and Figure H.**
>
> ---
>
> **Question 2.1**: It seems that sometimes the background scene can be blurry or unpleasant.
>
> **Answer**: Thank you for pointing is out. According to Equation (6) in the main text, the background regions are rendered using a global prompt. We believe that the blurry or unpleasant background you mentioned might result from the fact that many of our experimental results, such as Figures 3, 4, 5, and 6, are **directly taken from the outputs of the COCO-POS/COCO-MIG benchmark without any prompt optimization**, to ensure a fair comparison. Furthermore, **we did not use the SDXL refiner for post-processing**, which might also contribute to the observed issues.
>
> ---
>
> **Question 2.2:** Does that have something to do with the instance-wise rendering step?
>
> **Answer**: In real-world applications, users can **customize the global prompt based on their preferences**. For instance, to achieve a **bokeh effect** in the background, users can include prompts like "Bokeh"; to make the background **clearer and more detailed**, they can use prompts such as "Extreme detail description." This flexibility allows users to tailor the background rendering to specific aesthetic or functional requirements.
>
> ---
>
> **Question 3**: Are there any failure cases that can make analyses on the Layout-to-depth model?
>
> **Answer**: Thank you for your valuable question. In addition to common failure cases similar to those encountered in other layout-to-image models, such as **generating instances with nearly overlapping boxes** or **producing exceptionally small instances**, our Layout-to-depth model faces a significant challenge in generating specific shapes or poses. This issue stems from **the limitations of the SD1.5 backbone employed in the 3DIS's layout-to-depth model**, which hampers its ability to accurately render complex structures. This limitation is particularly noticeable in scenarios that involve text generating. For instance, generating a high-quality image of a strawberry cake with the text 'ICLR' written on it proves challenging; 3DIS may fail to produce correct scene depth maps, resulting in errors like misrepresenting the letter 'L' (as illustrated in **updated Appendix Section L, Figure K**). Future efforts could focus on **developing specialized datasets**, such as the proposed MARIO-10M dataset in TextDiffusers [1], to enhance the model's ability to handle complex structures. Enhancing the dataset specificity would markedly improve the robustness and versatility of 3DIS across a broader array of applications.
>
> > [1]: Textdiffuser: Diffusion models as text painters, 2023

---

> ### Author Response · Authors · 2024-11-19
>
> **Question 4:** How long will it take to generate a single image?
>
> **Answer**: Good question! We have added detailed runtime statistics in the updated **Appendix's Section A**. For **complex layouts with 10 instances**, the generation times on NVIDIA A100 GPUs are as follows:
>
> + **3DIS + SD1.5:** **11.07** seconds per 512x512 image.
> + **3DIS + SD2:** **17.08** seconds per 768x768 image.
> + **3DIS + SDXL:** **28.55** seconds per 1024x1024 image.
>
> **We also conducted a comparison with other adapter rendering methods**, including GLIGEN (Li et al., 2023b), Instance Diffusion (Wang et al., 2024), and MIGC (Zhou et al., 2024), using their default configurations on NVIDIA A100 GPUs. The results are summarized in the Table below:
>
> | Model                  | GLIGEN | InstanceDiff | MIGC | 3DIS (SD1.5) | 3DIS (SD2) | 3DIS (SDXL) |
> | ---------------------- | ------ | ------------ | ---- | ------------ | ---------- | ----------- |
> | **Inference Time (s)** | 12.75  | 42.48        | 6.81 | 11.07        | 17.08      | 28.55       |
> | **Resolution**         | 512    | 512          | 512  | 512          | 768        | 1024        |
>
>
> We can arrive at the following conclusion:
>
> + **3DIS demonstrates faster inference speeds with SD1.5**. Since the scene depth map generated by 3DIS does not require too high granularity, the speed of generating the scene depth map is very fast. The average inference time of **3DIS + SD1.5 is 11.07s**, even faster than GLIGEN and Instance Diffusion, which are based on the same SD1.5 base model.
> + **3DIS demonstrates acceptable inference speeds with SD2 and SDXL.** As we increase model capacity and image resolution, the inference time for 3DIS also rises. Rendering times are **17.08 seconds for SD2 and 28.55 seconds for SDXL, which we consider to be acceptable**. Additionally, our experiments show that using 3DIS with SDXL **even achieves faster processing speeds than InstanceDiffusion.** As discussed in Section 4.3, the performance of 3DIS + SDXL on COCO-MIG slightly surpasses that of InstanceDiffusion, demonstrating the practicality and efficiency of our 3DIS framework comprehensively.
>
> ---
>
> **Weakness 1:** This pipeline is largely dependent on the fine-tuned Layout-to-Depth generation model, and the final output uses the foundation diffusion model to render instances in a strictly confined region(depth). There might be inconsistencies between the layout-to-depth generation model and the renderer model, thus hindering the final performance.
>
>
>
> **Answer:** This is a very interesting perspective! While at first glance, decoupling a task into multiple stages might appear to introduce potential inconsistencies between stages, we believe this is not a concern in our framework. The reasons are as follows:
>
> + **State-of-the-art performance in COCO-POS Benchmark:** As shown in **Table 1** of the main text, the layout-to-depth generation model in 3DIS achieves SOTA results on the COCO-POS benchmark, significantly outperforming previous SOTA methods. Therefore, **our proposed approach of incorporating depth information through a staged decoupling training strategy enhances the model's position control capabilities** and proves to be highly practical.
> + **Improved attribute binding with depth maps**: As shown in **Table 2** of the main text, combining the depth maps generated by 3DIS with methods like GLIGEN or MIGC further improves their attribute binding performance. By providing accurate positional information, the model has a higher probability of rendering more fine-grained details. Thus, **for traditional multi-instance generation (MIG) tasks, 3DIS brings incremental benefits rather than hindering the final performance.**
> + **Potential for further optimization in the detail renderer:** The detail renderer, as a training-free module, offers substantial room for optimization. By incorporating other training-free algorithms, such as **InitialNO [1]**, we can achieve even more powerful rendering capabilities. Therefore, **for training-free high-resolution layout-to-image generation, 3DIS continues to demonstrate strong potential.**
>
> The decoupled design of 3DIS not only avoids inconsistencies but also enhances the system’s flexibility and scalability, making it a robust and practical framework for high-quality, position-controlled image generation.
>
> > [1] InitNO: Boosting text-to-image diffusion models via initial noise optimization. CVPR2024

---

> > ### Comment · Reviewer_Pjgj · 2024-11-22
> > **Response to authors**
> >
> > Thank you for the detailed explanation and analyses. Most of my concerns have been addressed.

---

> > > ### Author Response · Authors · 2024-11-22
> > >
> > > Thank you for your prompt response and positive feedback on our revisions. Please let us know if you have any further questions or suggestions.
> > >
> > > We also appreciate your valuable contributions throughout the review process!

---

> ### Author Response · Authors · 2024-11-19
>
> **Weakness 2**: The slightest artifacts in the depth map might cause great changes in the final generation results, like unwanted additional objects or background depth.
>
> **Answer**: Thank you for highlighting this concern. Indeed, within our 3DIS framework, numerous optimizations have been specifically designed to address the issue you have raised:
>
> - **The fine-tuning of LDM3D has significantly enhanced the quality of the generated depth maps**. **As illustrated in Appendix Section D and Figure D**, fine-tuning LDM3D with pyramid noise to emphasize low-frequency information enables a greater focus on global scene understanding and coarse attributes of instances, such as shape and pose. This ultimately improves the quality of the generated depth maps. **Additionally, we have incorporated the FID metric into the ablation study in the revised paper (Table 3), which demonstrates that fine-tuning LDM3D effectively enhances the quality of the final generated images**.
> - **The use of a low-pass filter effectively mitigates the impact of artifacts in depth maps**. **As demonstrated in Figure 5 of the initial paper**, we applied a low-pass filter to remove high-frequency noise from Controlnet's feature, directing the main generative network to focus more on the global layout information of Controlnet rather than detailed information. This approach significantly improves the quality of the generated images. **Moreover, we have added the FID metric to the ablation study in the revised paper (Table 4), and the results confirm that using a low-pass filter effectively improves the quality of the final generated images**.

---

### Official Review · Reviewer_LTry · 2024-11-04

**Soundness:** 2
**Presentation:** 3
**Contribution:** 2
**Rating:** 6
**Confidence:** 3

**Summary:**

The paper proposes a two-stage framework for multiple instance generation, which is 1)  generating a depth map conditioned on the layout map, 2) generating the image conditioned on the generated depth map with pretrained ControlNet.  Some techniques are also proposed to improve the results, such as using low-pass filter on the depthmap, using segment model for detail renderer.  The claimed improvements are on layout accuracy, instance accuracy and image quality.

**Strengths:**

1. The two stage framework for MIG seems to be novel, and the findings that depth map can be utilized as an intermediate product for  layout conditioned generation.
2. Some practical techniques are proposed, which may be useful in practical  applications.

**Weaknesses:**

1. Compared to one-stage methods, the proposed one can be computionally inefficient, the extra computation cost should be clearly evaluated and presented.
2. The paper heavily based on previous works, the novelty can be limited.
3. It seems that the key contribution to the improvement of MIOU is the depth map, especially as shown in Fig.3 and Fig. 4, the depth map already demonstrats advantage compared to MIGC results. Further discussion on why the depth map can better align with the layout can be added.

**Questions:**

1. How does the occlusion of the layout impact the depth map generation? Since there is no clue of which one is nearer. It is better to evaluate the generation quality of the depth map, and how the depth map impact the final generation.
2. The parameters of the low pass filter, and how to choose it are missing.

---

> ### Author Response · Authors · 2024-11-19
>
> Thank you for **recognizing the novelty and practicality of our work**! We have carefully considered many of the additional suggestions you provided.
>
>
>
> Below, we will address each of your questions one by one.
>
>
>
> **Question 1.1**: How does the occlusion of the layout impact the depth map generation? Since there is no clue of which one is nearer.
>
> **Answer:** Good question! **In the first stage of 3DIS, the layout-to-depth model is capable of automatically adjusting the depth order of instances without requiring explicit specifications**. Based on our experiments, 3DIS effectively generates scene depth maps under various conditions, including both complete and partial overlap of bounding boxes. **Detailed analysis and visual examples can be found in the updated appendix's Section B**.
>
> ---
>
> **Question 1.2**: It is better to evaluate the generation quality of the depth map, and how the depth map impact the final generation.
>
> **Answer**: Thank you for your feedback, we have also considered this issue. **In fact**, **the quality of the rendered image is strongly associated with the quality of the generated depth map. Therefore, the image quality metric (FID) in Table 1 also reflects the quality of the depth map.** our experiments indicate that **ControlNet-Depth is** **highly sensitive to the quality of the input depth map**. If the depth map contains noise, the generated images often exhibit severe distortions, such as **blotches** **or** **thread-like artifacts.** This sensitivity is illustrated in Appendix's Figure C. So we believe that the evaluation of depth map quality is **inherently reflected in the final image quality metrics**. Specifically, the quantitative results presented in Table 1 provide a comprehensive assessment of generation quality, and this strong correlation between depth map quality and final image quality ensures that **any deficiencies in depth map generation are already effectively captured in these evaluations**.
>
> ---
>
> **Question 2**: The parameters of the low pass filter, and how to choose it are missing.
>
> **Answer**: Thank you for pointing this out! We have clarified the parameters and rationale for the low-pass filter in the updated appendix's Section H. Specifically, the low-pass filter is applied to the ControlNet signals to emphasize low-frequency scene information while reducing high-frequency noise.
>
> **The parameters are defined as follows:**
>
> + **Frequency Mask Scale:** We use a scale of 0.5, which attenuates high frequencies outside a central region extending to H/4 and W/4 from the center of the frequency spectrum, where H and W represent the height and width of the residual vectors injected from the ControlNet.
> + **Choice of Parameter**: This parameter was selected through empirical testing, which was optimized through visualizing the generated outputs and iteratively refining the mask to achieve the best balance between reducing high-frequency noise and preserving essential object details.

---

> ### Author Response · Authors · 2024-11-19
>
> **Weakness 1**: Compared to one-stage methods, the proposed one can be computionally inefficient, the extra computation cost should be clearly evaluated and presented.
>
> **Answer**: We greatly appreciate your valuable feedback! In our experiment, we found that even with multiple stages of generation, 3DIS **can still maintain a good efficiency**.
>
> **Firstly, We conducted an efficiency analysis of 3DIS.** The 3DIS framework generates high-resolution images in three sequential stages **1)The Layout-to-Depth Model**, which creates a coarse-grained scene depth map; **2) The Segmentation Model**, which extracts the precise shape of each instance from the scene depth map; **3) The Detail Renderer**, which uses various foundational models (SD2, SDXL, etc.) to produce the final high-resolution image. We evaluated the inference efficiency of these stages using an NVIDIA A100 GPU. Our test involved a layout with 10 instances, and we assessed the inference time for each stage over 50 runs to calculate an average time:
>
> + **Layout-to-Depth Model**: The average time to generate a depth map is **5.66 seconds.**
> + **Segmentation Model**: We utilize the SAM model to segment the generated scene depth maps and get refined layouts. The refinement process by SAM takes **0.14 seconds.**
> + **Detail Renderer**: We use the EulerDiscreteScheduler for 50 steps.
>   - The time for the SD1.5 model to render a 512 x 512 image is **5.27 seconds**,
>   - the time for the SD2 model to render a 768 x 768 image is **11.28 seconds**,
>   - and the time for the SDXL model to render a 1024 x 1024 image is **22.75 seconds**.
>
> **Secondly, We also conducted a camparison experiments to test other adapter rendering methods**, including **GLIGEN (Li et al., 2023b)**, **Instance Diffusion (Wang et al., 2024)**, and **MIGC (Zhou et al., 2024)**, using their default configurations on NVIDIA A100 GPUs. The results are summarized in Table below:
>
> | Model                  | GLIGEN | InstanceDiff | MIGC | 3DIS (SD1.5) | 3DIS (SD2) | 3DIS (SDXL) |
> | ---------------------- | ------ | ------------ | ---- | ------------ | ---------- | ----------- |
> | **Inference Time (s)** | 12.75  | 42.48        | 6.81 | 11.07        | 17.08      | 28.55       |
> | **Resolution**         | 512    | 512          | 512  | 512          | 768        | 1024        |
>
>
> We can arrive at the following conclusion:
>
> + **3DIS demonstrates faster inference speeds with SD1.5**. Since the scene depth map generated by 3DIS does not require too high granularity, the speed of generating the scene depth map is very fast. The average inference time of 3DIS + SD1.5 is 11.07s, even faster than GLIGEN and Instance Diffusion, which are based on the same SD1.5 base model.
> + **3DIS demonstrates acceptable inference speeds with SD2 and SDXL.** As we increase model capacity and image resolution, the inference time for 3DIS also rises. Rendering times are **17.08** seconds for SD2 and **28.55** seconds for SDXL, which we consider to be acceptable. Additionally, our experiments show that using 3DIS with SDXL even achieves faster processing speeds than InstanceDiffusion. As discussed in Section 4.3, the performance of 3DIS + SDXL on COCO-MIG slightly surpasses that of InstanceDiffusion, demonstrating the practicality and efficiency of our 3DIS framework comprehensively.

---

> > ### Comment · Reviewer_LTry · 2024-11-22
> >
> > Thanks for the response.  My main concerns have been addressed. I would suggest the authors include the rebuttal points into the paper, especially how the depth map helps the generation.

---

> > > ### Author Response · Authors · 2024-11-22
> > >
> > > **Thank you for your suggestions, which have greatly contributed to the completeness of our work**.
> > >
> > > The **rebuttal points** mentioned above **have been incorporated** into the revised paper as follows:
> > >
> > > - **Results with occlusion layout**: These have been added to **Section B of the appendix**.
> > > - **Parameters of the low pass filter**: These details are included in **Section H** of the appendix.
> > > - **Inference efficiency analysis**: This analysis can be found in **Section A** of the appendix.
> > > - **Emphasizing the contribution**: This discussion has been expanded in **Section J** of the appendix.
> > > - **Analyzing the benefits of using a depth map**: This analysis is detailed in **lines 371-374 of the revised manuscript**.

---

> ### Author Response · Authors · 2024-11-19
>
> **Weakness 2:** The paper heavily based on previous works, the novelty can be limited.
>
> **Answer**: Thank you for your feedback! **While our proposed framework does **based on previous works**, we would like to emphasize its significant contributions to Multi-Instance Generation (MIG) and the broader AIGC field with the following points**:
>
> + **Motivation**: Previous layout-adapter methods have only released weights for SD1.5, **necessitating retraining for deployment on more powerful models like SD2 and SDXL**, which is both time-consuming and burdensome. Our 3DIS method divides MIG into two parts: scene depth construction and detail rendering. For scene depth construction, **we only train the layout adapter once for scene depth map generation**, focusing primarily on coarse-grained semantics, which **is adequately handled by the SD1.5 model.** **For detail rendering, 3DIS employs various stronger models and their widely pre-trained ControlNet in a training-free manner**, allowing users to benefit from the enhanced performance of increasingly powerful models.
> + **Technology**: Our 3DIS method restructures Multi-Instance Generation into two phases: constructing a scene depth map and training-free detail rendering. **This process differs significantly from previous approaches and has two notable features**: 1) **Generating a scene depth map rather than an RGB image** in the first stage allows the layout adapter to focus on coarse-grained attributes, **effectively improving its spatial control capabilities and handling overlapping scenarios** with added depth knowledge. 2) **The training-free detail rendering method enables users to utilize various foundational models** and their widely available pre-trained ControlNet **for rendering details directly**.
> + **Experiment Results:** Our experiments show that our method **surpasses previous approaches in location control** and allows the use of various foundation models for rendering without additional training costs, **resulting in markedly superior outcomes in detail rendering.**
> + **Success of Similar Work**: In recent years, we have seen many powerful pre-trained models. For most researchers, training a complete model from scratch is often impractical due to resource constraints. **Therefore, leveraging these pre-trained models to construct a novel framework that achieves new outcomes is both a meaningful and practical approach.** Many recent high-impact works [1, 2, 3] have adopted this strategy, underscoring its validity and relevance.
>
> > [1] Mastering Text-to-Image Diffusion: Recaptioning, Planning, and Generating with Multimodal LLMs. ICML2024
> >
> > [2] Layoutgpt: Compositional visual planning and generation with large language models. NIPS 2023
> >
> > [3] MultiDiffusion: Fusing Diffusion Paths for Controlled Image Generation. ICML 2023

---

> ### Author Response · Authors · 2024-11-19
>
> **Weakness 3:** It seems that the key contribution to the improvement of MIOU is the depth map, especially as shown in Fig.3 and Fig. 4, the depth map already demonstrats advantage compared to MIGC results. Further discussion on why the depth map can better align with the layout can be added.
>
> **Answer**: Thank you for your valuable feedback. We will include further discussion about this in the revised version. Overall, the improvement in the MIoU can be summarized in two main points:
>
> + **Emphasis on coarse-grained low-frequency information in depth maps enhances the generation of plausible scenes.** Unlike direct RGB image generation, depth map generation does not require the model to recover fine-grained details. This shifts the model's focus towards constructing global scenes and controlling coarse-grained attributes such as instance orientation and shape.
> + **The inherent properties of depth maps enable the network to better handle overlapping scenarios.** Let's take an example: as illustrated in Figure 3 of the initial paper, the RGB values of two pizzas generated by MIGC are very similar, which ultimately leads to a conjoined appearance in the generated results. In contrast, the depth maps generated by 3DIS show a significant difference in depth values between two pixels, which aids in preventing the confluence of these elements.

---

### Official Review · Reviewer_sPv1 · 2024-11-04

**Soundness:** 3
**Presentation:** 3
**Contribution:** 4
**Rating:** 8
**Confidence:** 4

**Summary:**

The paper introduces Depth-Driven Decoupled Instance Synthesis (3DIS), a novel framework aimed at enhancing multi-instance generation (MIG) in text-to-image generation. 3DIS addresses the limitations of existing MIG methods by decoupling the generation process into two distinct stages: generating a coarse scene depth map for accurate instance positioning and rendering fine-grained attributes without additional training. This two-stage approach allows for greater control over both layout and attribute details, leading to improved scene composition and integration with various foundational models. The extensive experimental results demonstrate that 3DIS significantly outperforms existing techniques in layout accuracy and fine-grained attribute rendering across established benchmarks.

**Strengths:**

- **Innovative Framework:** The 3DIS framework effectively decouples the multi-instance generation process, allowing for improved accuracy in scene composition and instance positioning.

- **Training-Free Approach:** The ability to render fine-grained attributes without additional training is a significant advantage, making the model more accessible and easier to integrate with existing systems.

- **Robust Performance:** Extensive experiments on benchmarks such as COCO-Position and COCO-MIG show that 3DIS consistently outperforms state-of-the-art methods in both layout precision and attribute rendering, indicating its effectiveness in practical applications.

**Weaknesses:**

- **Limited Dataset Scope**: The experiments relied on the LAION-art dataset and COCO benchmarks, which may not fully represent the diversity of real-world images or scenarios. Expanding the evaluation to include a broader range of datasets could provide a more comprehensive assessment of the model's generalizability.

- **Evaluation Metrics Limitations**: The selected evaluation metrics, while informative, may not capture all aspects of image quality or user satisfaction. Additional metrics could enhance the evaluation of the generated images' visual appeal and usability.

**Questions:**

1. How does the proposed 3DIS framework handle the challenges of noisy annotations in the input data?

2. The training procedure relied on the LAION-art dataset, which may not fully represent the diversity of real-world images or scenarios.

3. What strategies could be employed further to improve the robustness of the generated depth maps?

4. Given the dependence on pre-trained models for the training-free detail rendering process, how might variability in these models' performance impact the overall effectiveness of the 3DIS framework?

---

> ### Author Response · Authors · 2024-11-19
>
> We sincerely **appreciate your recognition of the innovative framework, strong flexibility, and robust performance of our 3DIS approach**. Thank you very much for your valuable comments!
>
> We will address all the concerns point by point.
>
> **Question 1**: How does the proposed 3DIS framework handle the challenges of noisy annotations in the input data?
>
> **Answer:** Thank you for raising this important question. We trained our **text-to-depth model using the LAION-art dataset** and our **layout-to-depth adapter using the COCO dataset**. **We will provide a detailed explanation** of how we address the challenges associated with noisy annotations in these datasets:
>
> + **LAION-art dataset**: The dataset contains numerous Web images along with their corresponding text captions. However, **these captions often include significant noise**, such as multilingual elements and prevalent advertising language. To address this issue, **we utilized the BLIP2 model to regenerate text** based on the images from the LAION-art dataset. **BLIP2 possesses robust image comprehension capabilities and is able to produce accurate captions.** We conducted a random sampling of 100 captions generated by the BLIP model for inspection. The BLIP-generated captions can precisely capture the key information of the image. Although the model has limitations in describing certain fine-grained attributes, these details are not crucial for depth map generation, where fine-grained specifics are less significant. **For a detailed explanation and specific examples, please refer to the updated appendix's Section G.**
> + **COCO dataset**: This dataset comprises numerous real-world images accompanied by **manually labeled textual annotations**, which are generally free from noise. However, **noise primarily arises during the instance description parsing process**. Initially, we employ the Stanza syntax parser to extract instance descriptions from the text annotations, followed by using Grounding-DINO to determine the positional annotations for each instance. During this phase, **the Stanza parser may incorrectly identify certain phrases as instances**, such as 'top', 'half', 'a group', and 'a couple'. To improve the accuracy of the instance descriptions, **we have compiled a filter list of these commonly misidentified phrases and actively filtered them out**.
>
> ---
>
> **Question 2**: What strategies could be employed to further improve the robustness of the generated depth maps?
>
> **Answer:** Thank you for this insightful query. Besides the techniques mentioned in the initial paper, we believe there are several additional approaches that could significantly enhance the robustness of the generated depth maps:
>
> + **Utilizing More Data**: Training our layout-to-depth model with an expanded dataset can enhance its robustness. For instance, incorporating **a broader range of vocabularies and corresponding concepts can help the model more accurately generate shapes that align with specific vocabularies**, thus improving precision and reducing errors in depth map generation.
> + **Implementing Negative Prompts**: Appropriately setting negative prompts can lead to higher-quality depth maps. We have found that **including terms such as "messy lines, weird lines, distorted lines" in negative prompts effectively prevents artifacts** in the depth maps. This approach guides the model to avoid generating unwanted features and focus on producing more accurate and realistic depth interpretations.
> + **Utilizing Stronger Models**: The generation of scene depth maps primarily focuses on coarse-grained, global scene understanding, rather than fine-grained instance attributes. Therefore, **using SD1.5 can adequately meet basic requirements**. **However, to achieve more robust performance, it is entirely feasible to train a layout-to-depth model based on stronger models.** Moreover, within our 3DIS framework, regardless of the underlying model used for the layout-to-depth module, it can seamlessly integrate in a training-free manner with most of the popular foundational models for rendering.

---

> ### Author Response · Authors · 2024-11-19
>
> **Question 2.2 (original)**: What strategies could be employed to further improve the robustness of the instance attributes?
>
> **Answer:** Thank you for your question regarding the enhancement of instance attribute robustness in our framework. We categorize instance attributes primarily into two types: 1) Coarse-grained instance attributes, such as shape and pose. 2) Fine-grained instance attributes, such as color, texture, and material. Approaches to further improve the robustness of instance attributes include:
>
> + **Coarse-grained Instance Attributes**: These attributes include shape and pose, which are determined during the scene depth maps generation stage. We believe that **introducing a broader range of data, including specific concepts of shape or pose**, can enhance the robustness of controlling these coarse-grained attributes. Additionally, we have updated the appendix to include examples of 3DIS effectively managing attributes like shape and pose.
> + **Fine-grained Instance Attributes:** These attributes include color, texture, and material, which are determined during the detail rendering stage. At this stage, **we utilize pre-trained models for rendering, which generally exhibit strong robustness due to their extensive training on diverse datasets**. However, to further improve the success rate of detail rendering, we propose that future improvements could involve incorporating training-free algorithms. One such method is **initial noise optimization**, such as InitNO [1], which can significantly enhance the rendering and control of fine details, ensuring more accurate and visually appealing outputs.
>
> > [1] InitNO: Boosting text-to-image diffusion models via initial noise optimization. CVPR 2024
>
> ****
>
> **Question 3**: Given the dependence on pretrained models for the training-free detail rendering process, how might variability in these models' performance impact the overall effectiveness of the 3DIS framework in real-world applications?
>
> **Answer:** Thank you for your question. In the training-free detail rendering process, we primarily utilize **three kinds of pretrained models: foundational generation models, segmentation models, and ControlNet models**.  We will discuss how the variability in these models' performance impacts the overall effectiveness of the 3DIS framework in real-world applications:
>
> + **Pretrained foundational generation models:** **1) stronger foundational models provide enhanced rendering capabilities.** For example, as illustrated in Table 2 of the initial paper, SDXL and SD2 show superior detail rendering ability compared to SD1.5. **2) different style weights can be applied depending on the scene**, allowing users to select models based on their creative preferences and requirements, such as using realistic style weights for data synthesis or anime style weights for artistic rendering, as shown in Figure 7 of the initial paper. **3) the choice of foundational models affects the rendering time**, as detailed in the appendix's Section A.
> + **Pretrained segmentation models:** SAM is used to accurately segment each instance, which is crucial for precise detail rendering. Figure 6 of the initial paper shows that precise instance location is key, especially in layouts with overlapping elements, where **incorrect segmentation could lead to improper attribute rendering**. However, **SAM, being a mature and extensively tested model, rarely encounters segmentation errors.** As segmentation technology advances, **3DIS can seamlessly integrate newer, more powerful models to improve instance positioning accuracy.**
> + **Pretrained ControlNet models**: ControlNet-Depth incorporates scene layout information to guide high-resolution generation models. **If ControlNet fails to align the generated image with the scene depth map, it could result in instance loss.** However, as observed, **ControlNet, frequently used within the AI art community, seldom experiences such errors.**
>
> It's noted that one of the core strengths of the 3DIS framework is its **dynamic, evolutionary, and adaptable nature**. This design allows for the seamless integration of more advanced pretrained models or alternative approaches as they become available, ensuring continuous improvement and resilience.

---

> ### Author Response · Authors · 2024-11-19
>
> **Weakness 1**: Limited Dataset Scope: The experiments relied on the LAION-art dataset and COCO benchmarks, which may not fully represent the diversity of real-world images or scenarios. Expanding the evaluation to include a broader range of datasets could provide a more comprehensive assessment of the model's generalizability.
>
> **Answer:** Thank you for your valuable feedback. Currently, Multi-Instance Generation (MIG) methods are primarily evaluated using benchmarks based on the LAION or COCO datasets [1, 2, 3], and **we have followed this established approach**. **I greatly appreciate your perspective and agree that it is essential.** **In future work, we plan to consider the diversity of real-world images and scenarios to develop a more comprehensive benchmark.** Additionally, to provide a more thorough assessment, **we have conducted an extra user study (please see the revised appendix's Section F)**. The results clearly indicate that our method is favored by users over previous state-of-the-art approaches.
>
> > [1] GLIGEN: Open-Set Grounded Text-to-Image Generation.
> >
> > [2] InstanceDiffusion: Instance-level Control for Image Generation.
> >
> > [3] MIGC: Multi-Instance Generation Controller for Text-to-Image Synthesis.
>
> ---
>
> **Weakness 2**: Noise in Input Data: The reliance on generated image captions from potentially noisy text descriptions could impact the quality of the depth maps and, consequently, the overall performance of the model. Addressing the challenges of noisy annotations is crucial for improving robustness.
>
> **Answer:** Thank you for highlighting this concern. **Indeed, the image captions generated by our system are of high quality.** As **mentioned in the answer to Question 1.1**, we have generated image captions only on the LAION-art dataset during the data processing phase. For this task, we employed the BLIP2 model, which **requires only the image input** and **does not rely on the original, potentially noisy text descriptions**. We have randomly **sampled several generated results and presented them in the appendix**. These results demonstrate that the captions produced by BLIP2 accurately convey the general information of the images. Although occasionally, the precision in fine-grained descriptions (such as color) may not be perfect, this level of detail is not crucial for generating coarse-grained depth maps. Therefore, overall, the quality of our generated captions is with no significant noise.
>
>
> ---
>
> **Weakness 3 (original)**: Evaluation Metrics Limitations: The selected evaluation metrics, while informative, may not capture all aspects of image quality or user satisfaction. Additional metrics could enhance the evaluation of the generated images' visual appeal and usability.
>
> **Answer:** Thank you for your feedback, which has allowed us to make our evaluation criteria more comprehensive. **We have included the FID metric in our paper's tables to assess image quality and conducted a user study to evaluate user preferences**:
>
> + **FID Metric for Image Quality:** In Table 1 of the initial paper, we used the FID metric to assess the quality of images generated by various models. The results show that the image quality produced by 3DIS surpasses previous methods. This is further corroborated by the visual comparisons in Figures 3 and 4 of the initial paper. Additionally, to validate the impact of our proposed modules on image quality, we have incorporated the FID metric into the ablation study in the revised paper (i.e., Tables 3 and 4) to demonstrate that our modules do not compromise image quality and, in some cases, even enhance it.
> + **User Study for User Satisfaction:** We have included the results of a user study in the revised appendix’s Section F. Specifically, we invited 30 individuals and randomly selected 30 results from the COCO-MIG benchmark for InstanceDiffusion, MIGC, and 3DIS for them to rank according to their preferences. The results indicate that the majority of users prefer the outcomes produced by 3DIS.

---

### Official Review · Reviewer_m1R4 · 2024-11-07

**Soundness:** 4
**Presentation:** 3
**Contribution:** 3
**Rating:** 8
**Confidence:** 3

**Summary:**

This work focuses on controllable image generation and points out the unified adapter challenge of multi-instance generation (MIG) methods: current MIG methods uses a single adapter to simultaneously handle instance positioning and attribute rendering. Such a unified structure complicates the development of detail renderers because it requires a large number of high-quality instance-level annotations. To this end, this work proposes a two-stage generation paradigm: (1) generating a coarse-grained depth map from layout; (2) rendering fine-grained instance details from depth map. This design enables the MIG adapter to be seamlessly integrated into various foundational models such as SD2 and SDXL without specific training. Extensive experiments demonstrate the effectiveness and flexibility of the proposed method.

**Strengths:**

1. This paper has a very clear motivation and solves the pointed problems very well. Although the idea of ​​using depth map as a coarse-grained scene guidance is simple, it effectively solves the limited adaptability challenge in existing MIG methods. Extensive qualitative and quantitative results (e.g., Table 2, Figure 3, and Figure 4) demonstrate that the proposed method has strong flexibility and can be applied to a variety of foundational models.
2. The presented results are promising, achieving state-of-the-art performance on instance attributes and locations. In particular, the proposed method can be further combined with other controllable image generation methods such as GLIGEN and MIGC to improve their performance for multi-instance generation..
3. The paper is well written and clearly structured.

**Weaknesses:**

1. The proposed framework depends on many existing models. For example, a) the text-to-depth model is obtained by fine-tuning LDM3D; b) a pretrained depth-conditioned ControlNet is required for depth layout injection; c) the detailed renderer relies on SAM to segment instances from depth. These dependencies weaken the originality of the paper and may compromise the robustness of the proposed framework.
2. The paper does not discuss depth ambiguity when multiple bounding boxes overlap. For example, given two partially overlapping instances, how can the model tell which is in front and which is behind?

**Questions:**

1. How does the layout adapter connect to the text-to-depth model? Does it work like ControlNet?
2. Now that depth information is introduced, can this method control the front-back relationship of overlapping instances?

---

> ### Author Response · Authors · 2024-11-19
>
> We sincerely **appreciate your recognition of the good performance and strong flexibility of our 3DIS approach, as well as your acknowledgment of our writing**. Thank you very much for your valuable comments!
>
> We will address all the concerns point by point.
>
>
>
>
>
> **Question 1.1:** How does the layout adapter connect to the text-to-depth model?
>
> **Answer:** The layout adapter connects to the text-to-depth model **through the incorporation of novel trainable attention layers situated within the cross-attention strata of the U-Net architecture.** This specific method of connection is delineated in Lines 165-169 of the initial paper. **We greatly appreciate your feedback and will enhance its clarity and comprehensiveness in the forthcoming revision of our paper.**
>
>
> ---
>
>
> **Question 1.2:** Does the layout adapter work like ControlNet?
>
> **Answer:** Thank you for this question. **The layout adapter differs from ControlNet.** The two main differences between them can be summarized as follows:
>
> + **Differences in Architecture Modification**: 1) **ControlNet** employs an auxiliary network, essentially adding an entire parallel network that operates alongside the main model. 2) The **layout adapter** (like GLIGEN, MIGC, and InstanceDiffusion) makes more localized adjustments by altering only the cross-attention layers within the UNet’s existing structure.
> + **Differences in Injecting Information**: 1) **ControlNet** incorporates image conditions through its auxiliary network, which processes these conditions separately and then merges the results with the main model’s output. 2) The **layout adapter**, however, embeds layout information directly into the generation process by introducing new, trainable attention layers within the existing cross-attention structure of the U-Net.
>
> ---
>
>
>
> **Question 2:**  Now that depth information is introduced, can this method control the front-back relationship of overlapping instances?
>
> **Answer:** Good Question! **Yes, without requiring additional training, our 3DIS method can control the front-back relationships between instances**. The corresponding results are provided in the revised appendix; **please refer to the appendix's Section B and Figure A for details.**

---

> ### Author Response · Authors · 2024-11-19
>
> **Weakness 1:**  The proposed framework depends on many existing models.
>
> **Answer:** Thank you for your feedback! **While our proposed framework does depend on many existing models, we would like to emphasize its significant contributions to Multi-Instance Generation (MIG) and the broader AIGC field with the following points**:
>
> + **Motivation**: **Previous layout-adapter methods have only released weights for SD1.5**, **necessitating retraining for deployment on more powerful models like SD2 and SDXL**, which is both time-consuming and burdensome. Our 3DIS method divides MIG into two parts: scene depth construction and detail rendering. For scene depth construction, **we only train the layout adapter once for scene depth map generation**, focusing primarily on coarse-grained semantics, which is adequately handled by the SD1.5 model. For detail rendering, 3DIS employs various stronger models and their widely pre-trained ControlNet in a training-free manner, **allowing users to benefit from the enhanced performance of increasingly powerful models.**
> + **Technology:** Our 3DIS method restructures Multi-Instance Generation into two phases: constructing a scene depth map and training-free detail rendering. **This process differs significantly from previous approaches and has two notable features**: 1) **Generating a scene depth map rather than an RGB image** in the first stage allows the layout adapter to focus on coarse-grained attributes, **effectively improving its spatial control capabilities and handling overlapping scenarios** with added depth knowledge. 2) **The training-free detail rendering method enables users to utilize various foundational models** and their widely available pre-trained ControlNet for rendering details directly.
> + **Experiment Results:**  Our experiments show that our method **surpasses previous approaches in location control** and allows the use of various foundation models for rendering without additional training costs, **resulting in markedly superior outcomes in detail rendering.**
> + **Success of Similar Work:** In recent years, we have seen many powerful pre-trained models. For most researchers, training a complete model from scratch is often impractical due to resource constraints. **Therefore, leveraging these pre-trained models to construct a novel framework that achieves new outcomes is both a meaningful and practical approach.** Many recent high-impact works [1, 2, 3] have adopted this strategy, underscoring its validity and relevance.
>
> > [1] Mastering Text-to-Image Diffusion: Recaptioning, Planning, and Generating with Multimodal LLMs. ICML2024
> >
> > [2] Layoutgpt: Compositional visual planning and generation with large language models. NIPS 2023
> >
> > [3] MultiDiffusion: Fusing Diffusion Paths for Controlled Image Generation. ICML 2023
>
>
> ---
>
>
> **Weakness 2:**  The paper does not discuss depth ambiguity when multiple bounding boxes overlap. For example, given two partially overlapping instances, how can the model tell which is in front and which is behind?
>
> **Answer:** We greatly appreciate your valuable feedback. In response, **we have included a detailed discussion in the appendix** to enhance the completeness of our work. **For further details, please see the appendix's Section B and Figure B.** Our 3DIS method can generate various scene depth maps from overlapping layouts with depth ambiguity by using different seeds. **Given two partially overlapping instances, this process creates different arrangements of instances on depth order, consistently aligning with the original layout.**

---

> > ### Comment · Reviewer_m1R4 · 2024-11-21
> >
> > Thanks the authors for the detailed responses. The mentioned training-free strategy to control the layout order and its results are interesting. My concerns have been addressed, so I will keep my original rating.

---

> > > ### Author Response · Authors · 2024-11-21
> > >
> > > Thank you for your prompt response and positive feedback on our revisions. Please let us know if you have any further questions or suggestions.
> > >
> > > We also appreciate your valuable contributions throughout the review process!

---

### Meta-Review · Area_Chair_Ewt5 · 2024-12-19

**Metareview:**

- **Strengths**:
  - Clear and novel decoupling of the MIG pipeline.
  - Demonstrated improved spatial control, flexibility, and integration with pre-trained models.
  - Robust validation through metrics, qualitative examples, and user studies.

- **Weaknesses**:
  - Dependency on existing models limits perceived originality.
  - Persistent concerns about generalizability and scalability.
  - Incremental improvements over prior art.

The overall decision was to **accept (spotlight)**, with reviewers acknowledging the significant technical contributions and improvements made during the rebuttal period. The work is deemed impactful, with potential for further refinements.

**Additional Comments On Reviewer Discussion:**

## Points Raised by Reviewers

1. **Framework Dependence**:
   - Heavy reliance on existing models like SAM, ControlNet, and SD for rendering, raising concerns about originality and robustness.
   - The potential for inconsistencies between the layout-to-depth generation model and the renderer model.

2. **Evaluation Metrics and Analysis**:
   - Lack of efficiency analysis for multi-step pipeline.
   - Missing ablation studies for image quality metrics like FID.
   - Limited user study to assess user preferences.

3. **Depth Map Artifacts**:
   - Potential issues with depth map artifacts causing unwanted elements in the final generation.
   - Sensitivity of the final image quality to the depth map accuracy.

4. **Limited Dataset Scope**:
   - Evaluation is primarily based on COCO and LAION-art datasets, limiting generalizability to real-world applications.

5. **Visual Quality Concerns**:
   - Issues with blurry or unpleasant background scenes.
   - Suggestions to enhance fine-grained instance details in visual outputs.

6. **Novelty of the Framework**:
   - Incremental improvements over previous works, with reliance on existing techniques.

## Author Responses and Revisions

1. **Framework Contributions**:
   - Clarified the novelty of decoupling the MIG process into layout-to-depth generation and training-free rendering.
   - Emphasized improved spatial control and adaptability to multiple foundational models.

2. **Efficiency Analysis**:
   - Provided detailed runtime benchmarks for different models (SD1.5, SD2, and SDXL) on NVIDIA A100 GPUs.
   - Compared 3DIS to other methods, demonstrating competitive inference times.

3. **Depth Map Quality**:
   - A low-pass filter was introduced to reduce noise and artifacts in depth maps.
   - Fine-tuned LDM3D to emphasize coarse-grained attributes and enhance global scene understanding.

4. **Expanded Validation**:
   - Conducted user studies with 30 participants to evaluate user satisfaction, showing a preference for 3DIS over other methods.
   - FID metrics were included in ablation studies to assess image quality.

5. **Addressing Dataset Limitations**:
   - Acknowledged dataset constraints and committed to expanding evaluation in future work.
   - Added diverse examples and failure cases in the appendix to illustrate robustness.

6. **Enhanced Visual Outputs**:
   - Demonstrated flexibility in adjusting global prompts to improve background aesthetics.
   - Failure cases and strategies for further robustness are discussed in the appendix updates.

## Final Decision Rationale

- **Strengths**:
  - Clear and novel decoupling of the MIG pipeline.
  - Demonstrated improved spatial control, flexibility, and integration with pre-trained models.
  - Robust validation through metrics, qualitative examples, and user studies.

- **Weaknesses**:
  - Dependency on existing models limits perceived originality.
  - Persistent concerns about generalizability and scalability.
  - Incremental improvements over prior art.

The overall decision was to **accept (spotlight)**, with reviewers acknowledging the significant technical contributions and improvements made during the rebuttal period. The work is deemed impactful, with potential for further refinements.

---

### Decision · Program_Chairs · 2025-01-22

Accept (Spotlight)